# THE BENEFITS OF MODEL-BASED GENERALIZATION IN REINFORCEMENT LEARNING

## ABSTRACT

Model-Based Reinforcement Learning (RL) is widely believed to have the potential to improve sample efficiency by allowing an agent to synthesize large amounts of imagined experience. Experience Replay (ER) can be considered a simple kind of model, which has proved extremely effective at improving the stability and efficiency of deep RL. In principle, a learned parametric model could improve on ER by generalizing from real experience to augment the dataset with additional plausible experience. However, owing to the many design choices involved in empirically successful algorithms, it can be very hard to establish where the benefits are actually coming from. Here, we provide theoretical and empirical insight into when, and how, we can expect data generated by a learned model to be useful. First, we provide a general theorem motivating how learning a model as an intermediate step can narrow down the set of possible value functions more than learning a value function directly from data using the Bellman equation. Second, we provide an illustrative example showing empirically how a similar effect occurs in a more concrete setting with neural network function approximation. Finally, we provide extensive experiments showing the benefit of model-based learning for online RL in environments with combinatorial complexity, but factored structure that allows a learned model to generalize. In these experiments, we take care to control for other factors in order to isolate, insofar as possible, the benefit of using experience generated by a learned model relative to ER alone.

Model-based reinforcement learning (RL) refers to the class of RL algorithms which learn a model of the world as an intermediate step to policy optimization. One important way such models can be used, which will be the focus of this work, is to generate imagined experience for training an agent's policy (Werbos, 1987; Munro, 1987; Jordan, 1988; Sutton, 1990; Schmidhuber, 1990). Experience Replay (ER) can be seen as a simple, nonparameteric, model (Lin, 1992; van Hasselt et al., 2019) where experienced interactions are directly stored, and later replayed, for use in a learning update.

ER already captures many of the benefits associated with a learned model as compared to model-free incremental online algorithms (i.e. model-free algorithms which perform a learning update using each transition only at the time it is experienced). In particular, ER allows value to be rapidly propagated from states to their predecessors along previously observed transitions, without the need to actually revisit a particular transition for each step of value propagation. Propagating value only at the time a transition is visited can make model-free incremental online algorithms extremely wasteful of data, particularly in environments where the reward signal is sparse.

As Lin (1992) and van Hasselt et al. (2019) have discussed, it is often not obvious why we'd expect experience generated by a learned model to improve upon ER, as a replay buffer is essentially a perfect model of the world insofar as the agent has observed it. This is especially true in the tabular case, where a model does not generalize from the observed transitions. It is also true for policy evaluation in the case where the value function and model are linear (Sutton et al., 2012). In this case, learning the least-squares linear model from the data, and then finding the TD(0) solution (Sutton, 1988) in the resulting linear MDP is identical to finding the TD(0) solution for the empirical MDP induced by the observed data. Hence, if we expect to obtain a sample efficiency benefit by using data generated by a learned model compared to ER, we should look beyond these cases.

Let's now consider a situation where a model-free agent using ER will likely fail to generalize from experience in a way that an intelligent agent should be able to. Imagine an agent has witnessed a

tree lying across a stream which allows it to cross in order to reach food on the other side. Imagine the same agent has previously pushed against a decaying tree and knocked it over, but in that case gained nothing by doing so. We may then expect that upon seeing a decaying tree standing near a stream with food on the other side, the agent would be able to synthesize these two experiences to decide to intentionally push the tree over to provide a bridge over the stream.

We argue that ER alone is unlikely to achieve the kind of generalization necessary to perform such extrapolation. To understand why, let's express the state as a combination of two abstract binary factors (tree fallen, food across stream). The agent has observed state (1,1), and also observed separately that from state (0,0) it can achieve state (1,0) through a certain action (call that action *push-tree*). Now, the agent observes state (0,1). An agent using ER can easily learn that (1,1) is valuable since it is followed by a food reward, but since it has never executed push-tree from (0,1) to reach that valuable state, there is no reason to expect it will assign that action an elevated value.

On the other hand, a learned model that generalizes appropriately could guess, perhaps based on some inductive bias toward factored dynamics[1], that taking action push-tree in (0,1) would lead to (1,1) in the next step. Having done so, we could then use the model in various ways to incorporate this information into the value function and/or policy. One common way of using the model, that we will focus on in this work, is to generate (potentially multi-step) rollouts of simulated experience and train a value function or policy on the resulting trajectories as if they were real.

Having discussed how learned model generalization can provide a benefit over ER, it's worth noting that parametric value functions also generalize. Why should model generalization be inherently better than value function generalization? This question was already raised in the work of Lin (1992), which first introduced ER for RL. The next section gives a partial answer as a theorem which shows how learning a model as an intermediate step can narrow the space of possible value functions more than learning a value function directly from the data using the Bellman equation.

After motivating the benefit of model-based generalization theoretically, we will present an intuitive case where learning a parametric model is empirically beneficial with NN function approximation. Subsequently, we will present extensive experiments, in environments with factored structure, which highlight the sample-efficiency benefits of model-based learning for online RL. We will also analyze an interesting instance we came across during these experiments where an agent using a learned model outperforms one using the perfect model due to smoothed reward and transition dynamics.

We use the MDP formalism throughout this paper. We refer the unfamiliar reader to Sutton & Barto (2020) for an accessible discussion of MDPs and their usage in RL.

## 1    UNDERSTANDING THE BENEFIT OF MODEL-BASED GENERALIZATION

We now present a simple theorem that provides at least part of the answer to the question of how model generalization can be considered more useful than value function generalization. We state the theorem informally here, and formally with proof in Appendix A.

**Theorem 1.** *Consider a class of deterministic, episodic, MDPs $\mathcal{M}$ with fixed reward function, and transition function belonging to some known hypothesis class. Let $H_Q$ be the associated class of optimal action-value functions for MDPs in $\mathcal{M}$. Now consider a dataset $D$ of transitions. Let $H_B(D)$ be the subclass of action-value functions in $H_Q$ which obey the Bellman optimality equation for the transitions in $D$ and let $H_M(D)$ be the subclass of optimal action-value function of MDPs in $\mathcal{M}$ which are consistent with $D$. Then the following are true:*

1. *$H_M(D) \subseteq H_B(D)$.*
2. *For some choices of $\mathcal{M}$ and $D$, $H_M(D) \subset H_B(D)$.*
3. *For a tabular transition function class, that is one that includes every possible mapping from state-action pairs to next states, $H_M(D)=H_B(D)$.*

Intuitively Theorem 1 states that, if we want to narrow down the possible optimal action-value functions from data, we can in general prune more if we narrow down the possible models first than if we only demand that the value functions obey the Bellman optimality equation with respect to the observed data. However, part 3 states that this is not the case for a tabular model class, but rather only

---

[1]Where the state consists of a set of state-variables such that the distribution of values of each variable at the next time-steps depends on only a subset of the variables in the current state.

for models which have some additional structure. In such cases, even if a model-based and model-free approach begin with the same hypothesis class, the model-free approach can fail to leverage this structure and ultimately lose value-relevant information from the data in the process. While the assumptions of Theorem 1 differ from the reality of training neural networks (NNs) with ER, this theorem provides intuition for why learning a parametric model can improve sample efficiency. In the next section, we will highlight a simple and intuitive case where similar intuition leads a learned model to be empirically beneficial with NN function approximation.

## 2 A SIMPLE CASE WHERE MODEL-BASED GENERALIZATION IS USEFUL

In this section, we present an illustrative example where learning a parametric model from data, and then learning an action-value function within that model, has a clear advantage over learning an action-value function directly from the data. We also show that using multi-step model rollouts has an advantage over single-step rollouts. To keep this example as simple and intuitive as possible, we consider an offline RL setting with hand-selected datasets with varying coverage of the dynamics.

The environment in this section consists of a 3x3 grid world with a goal in the top left corner and with arbitrary walls (the location of which is fixed within an episode). Reward is -1 at each step until the goal is reached, at which point the episode terminates. Actions consist of standing still, or moving in a cardinal direction. Observations are flat binary vectors consisting of one-hot-encodings of the agent's position and the goal position (though goal position is fixed here), and two binary vectors indicating, respectively, the presence and absence of walls in each cell.

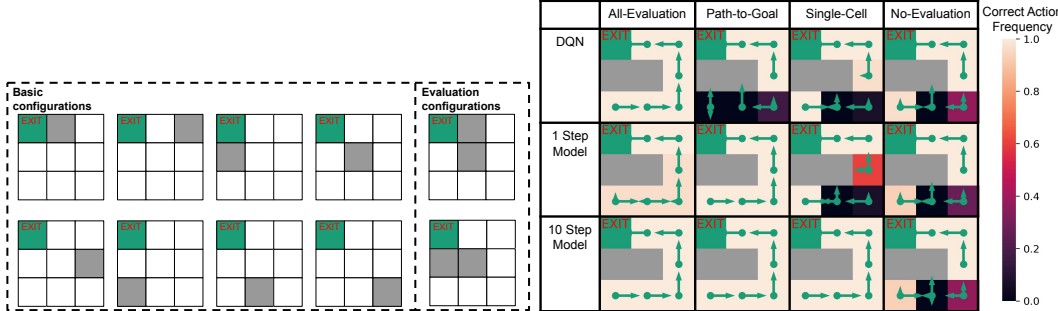

Figure 1: **Left**: Maze layouts included in the basic and evaluation sets. All datasets include every transition in each of the basic configurations, but different sets of transitions in the evaluation configurations. **Right**: Frequency (over 30 random seeds) of trained agents' greedy policy selecting the correct action in cells of the displayed maze with different evaluation set coverage during training. Length of green arrows indicate the frequency of greedy policies picking the corresponding action.

For explanatory purposes, we break the data in the training datasets provided to the agents into a *basic set* and an *evaluation set*. The basic set consists of all possible transitions with wall layouts having a single wall within one of the 8 non-goal cells. Every training dataset contains this entire basic set, in addition to some limited data from the evaluation set. The evaluation set consists of transitions with wall layouts of 2 walls in one of the configurations illustrated in Figure 1(left). We consider training datasets with 4 different levels of evaluation set data coverage:

- **All-Evaluation**: All possible transitions within the evaluation wall layouts.
- **Path-to-Goal**: Only evaluation layout transitions which follow the path to the goal.
- **Single-Cell**: Only the transitions starting from the cell furthest from the goal (with respect to the only open path).
- **No-Evaluation**: No transitions from the evaluation set.

We compare model-based and model-free algorithms in this setting. For the model-free approach, we use a Deep Q-Network (DQN; Mnih et al. (2015)) with ER. For the model-based approach, we use a simple feedforward NN model, and DQN trained on model generated transitions. The model takes an observation as input and outputs a predicted reward, Bernoulli termination probability and vector of Bernoulli probabilities that each feature is active in the next state. Note that this model cannot capture correlation between features in the next observation given the current observation. However, as the proposed environment has deterministic transitions, it is sufficient to capture the true dynamics. We train the model on transitions from the dataset and subsequently train the action-

value function on model rollouts initialized from states in the dataset. We trained one model-based agent with single-step rollouts and another with 10-step rollouts. We trained each agent for 1 million training steps and controlled the total number of (real or imagined) transitions used in each DQN update across all agents. See Appendix E for further detail on the experiment setup.

Each agent is evaluated by checking the frequency out of 30 independent runs with which the greedy action under the learned value function is optimal in each cell of an evaluation layout. We say the agent has failed if there exists any cell in which the majority of runs select the wrong greedy action. We next discuss how we expect each agent to perform with each level of evaluation set coverage.

**All-Evaluation**: We expect that all agents will succeed in the case where data is given for all transitions in the evaluation layouts. Here, a model-free agent has access to all the data needed to backup value from the goal and determine the value of each state-action pair, and a model-based agent has no opportunity to generate useful novel transitions which do not already appear in the dataset.

**Path-to-Goal**: We expect model-free DQN to fail in the Path-to-Goal case. Consider the lower evaluation wall layout in Figure 1(left). For the bottom middle cell, there is no data in the Path-to-Goal dataset for actions besides moving right. For every basic-set layout, moving right is worse than any action which remain in the same cell. We predict the model-free agent will incorrectly generalize from the basic set to conclude that the right action is also worse in this new configuration.

We expect both model-based agents to succeed with Path-to-Goal data. All missing transitions are one step away from the available data, but require a different action selection. We predict that the outcome of these missing actions can be learned by generalization from the basic set. As in the simple situation described in the introduction, this environment has significant factored structure. To determine the effect of an action, it suffices to look at the agent's current location and whether there is a wall in the cell it is attempting to enter. We predict the agent will be able to learn this basic structure from the training data, even in the absence of specific data about the case where there are two walls. Note that this hypothesis implies a nontrivial prediction about how this simple model will generalize. Factored structure is not hard-coded in the model, thus it is also plausible that the model predictions will be arbitrarily bad for the unobserved evaluation transitions.

**Single-cell**: We expect the single-step model to fail when only transitions from a single evaluation cell are included in the dataset. In this case, the model rollouts have no chance to reconstruct anything not already available explicitly in the dataset given all single step transitions from the far cell are included and this is likely to be insufficient for reasons already explained. However, the 10-step model has the potential to succeed. If the model generalizes as expected, it can start in the far cell available in the dataset and rollout a trajectory which discovers the full path to the goal.

**No-Evaluation**: Finally, all agents should fail in the case where there is no evaluation data available at all. The models will have no opportunity to provide the learned value function with example transitions from the evaluation layout given model rollouts are initialized with states from the dataset. Note that the situation may be different if we had used a generative start-state model to produce plausible states for the start of rollouts which need not explicitly appear in the dataset.[2] Using the model to plan for immediate action selection at evaluation time, as in model predictive control, would also help here, as the agent could directly plan a response for the previously unseen state.

In Figure 1(right), we observe that all the above predictions are confirmed. We reiterate that this is a nontrivial empirical result. It relies on the simple model, a feedforward NN, with no explicit bias toward factored solutions, generalizing in a particular way to state-action pairs that do not explicitly appear in the dataset. At least in this case, the model indeed seems to generalize in a way that provides a significant advantage over ER alone.[3] This experiment also straightforwardly illustrates how even a model with 1-step rollouts can be helpful, by sampling counterfactual actions or, in the case of stochastic environments, counterfactual chance outcomes. However, multi-step rollouts can succeed in situations where there is insufficient data for one-step rollouts to be helpful.

---

[2]However, this raises questions about how the start-state model should generalize, if trained to maximize data likelihood, it could overfit and only generate states from the training set, rendering it unhelpful.

[3]We also verified that the model predictions in this case are near perfect with respect to predicting the agent position after each state-action pair across all transitions in the evaluation configurations.

## 3 FAVORABLE ENVIRONMENTS FOR ONLINE MODEL-BASED LEARNING

We next describe three environments which exemplify properties that should make online learning with a parametric model particularly useful. We aim for the following environment characteristics:

1. The state-space should have a simple factored structure that we expect should be easy to learn for a model with reasonable generalization properties.
2. The return should depend sharply on the policy, such that a randomly behaving agent won't have much of a learning signal for policy improvement. This should make model generated experience more useful, as Bellman backups alone will tend to be mostly uninformative.
3. To allow the model-based agent to learn about the reward function, the agent should occasionally witness rewarding states even while behaving highly sub-optimally. Toward this, we include occasional random transitions to rewarding states in our subsequent experiments. The same is true for terminal states in environments where termination is desirable. This does not contradict the previous point, as an agent may be able to occasionally obtain some reward while behaving sub-optimally, but have difficulty obtaining more.

The last characteristic may seem contrived, but we argue that it is quite natural. For example, one can imagine an agent gathering edible plants for a long time before working out how to grow their own. It is often much harder to discover reward, without ever observing it, than to work out how to reconstruct rewarding circumstances using general knowledge of the transition dynamics. On a practical note, there is significant work on exploration with sparse (or no) reward (Schmidhuber, 1991a;b; Thrun, 1992; Schmidhuber, 2010; Amin et al., 2021) which is orthogonal to our focus. Hence, we mitigate the issue by making it easier to learn the reward function. We ablate these spontaneous transitions to rewarding states in Appendix I to test the impact of this choice.

In addition to the above, the environments we investigate allow scaling of problem complexity to test the limitations of different approaches. The environments are also Markov, use binary features, and are largely deterministic, so simple models can work well, though we also investigate more sophisticated latent-space models. Next, we describe the environments (see Appendix B for details).

ProcMaze        ButtonGrid        PanFlute

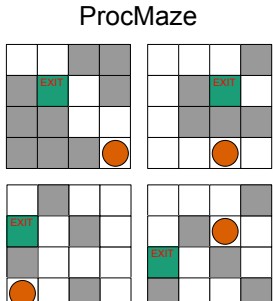 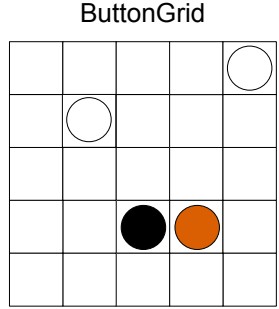 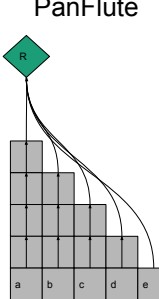

Figure 2: **Left**: Examples of states in the ProcMaze environment of size 4 with the agent shown in orange. **Middle**: An example state of the ButtonGrid environment with 3 buttons. The agent is shown in orange and the buttons in black (off) and white (on). **Right**: An instance of the PanFlute environment with 5 pipes. The agent directly activates cells (a,b,c,d,e) through its actions after which the activation propagates up the associated pipe, one step at a time, and dissipates at the end.

**ProcMaze** (Figure 2, left): Procedurally generated grid world mazes. The maze itself, along with the start state and goal state are randomized in each episode. Complexity is scaled by increasing the grid size. Negative reward is given for each step until the goal is reached.

**ButtonGrid** (Figure 2, middle): A 5 by 5 grid with randomly placed buttons. An agent can move around and, if it hits a button, will toggle it on or off. If all buttons are on, a reward is given and button locations are randomized. Random behavior will tend to randomly perturb the buttons, a precise policy is required to set them all to on. Complexity scales with the number of buttons.

**PanFlute** (Figure 2, right): A minimal example of an environment with combinatorial complexity of optimal behavior, but simple factored transition structure. PanFlute consists of $n$ *pipes* of cells where each pipe evolves independently. Each action directly activates the cell at the bottom of one pipe, after which the activation will propagate up the pipe, one step at a time, and dissipate after reaching

the end cell of the pipe. A reward is received if the cells at the end of all pipes are simultaneously active, which can only be achieved by choosing each of the $n$ actions in a certain order, a probability of $1/n^n$ under random behavior. We scale the complexity by changing the number of pipes $n$.

We expect ER alone to be of limited utility in each of these environments as each of them require precise control to obtain significantly more reward than random behavior, especially as the problem complexity is scaled up in each case. Since we always include random transitions to rewarding states, an agent can easily learn that these states are good, but until it reaches the rewarding state by it's own actions, it wont be able to learn much from the states which proceed rewarding states.

On the other hand, each environment has significant factored structure which a model-based agent can learn, and subsequently use to imagine many novel, plausible, states in its rollouts. In ProcMaze, an agent moving into a specific empty space will have the same effect regardless of the rest of the maze layout, and attempting to move into a wall will always block it. In ButtonGrid, the connectivity of the grid is independent of the button layout, and stepping on a specific button always has the same effect regardless of the layout of the rest of the buttons. In PanFlute, each action always has the same effect, and each pipe evolves according to dynamics which are unaffected by the other pipes. In appendix C, we present an experiment in an environment without factored structure and find that ER outperforms a learned model in this case, which helps to contextualize our other results.

## 4 BENEFICIAL MODEL-BASED GENERALIZATION FOR ONLINE RL

We now empirically evaluate the performance of model-based and model-free learning algorithms on the environments described in Section 3. We experiment with variants of each environment with a range of different complexities to test how different approaches scale to more complex environments.

All tested approaches use DQN for behavior learning, but vary in how training examples for DQN are generated. Our model-free approach draws transitions randomly from an ER buffer for training. We test several types of learned model. The first is the simple feedforward NN model introduced in Section 2. The second is a latent-space model (Schmidhuber, 1997; Watter et al., 2015; Ha & Schmidhuber, 2018) inspired by Dreamer (Hafner et al., 2019; 2020), but with two major differences to simplify the approach and reduce confounding factors in our experiments. In particular, as noted, we use DQN instead of actor-critic and, since our environments are Markov, we forgo the recurrent network of Dreamer and model single-step stochastic transitions with no memory. We experiment with Gaussian and Categorical latent variables. See Appendix D for further details of these models. Finally, as a strong baseline, we include a perfect model, which uses the ground truth environment dynamics instead of a learned model, but is otherwise the same as the simple-model agent.

As in Section 2, we control for the total number of updates, and the total number of (real or imagined) transitions used in each update. In particular, model-free DQN is updated on a batch of 320 transitions from the replay buffer while all model-based approaches use 32 model rollouts of length 10, beginning in a state from the replay buffer. In Appendix I, we perform an ablation in which 1-step rollouts are used instead and find that longer rollouts are generally helpful. In each update, the model is trained using the same batch of transitions which initialize the rollouts. All agents use a softmax behavior policy, and are evaluated under the greedy policy. All Q-networks are trained on discounted 1-step TD-error, with real or imagined transitions, using a discount factor of $0.9$.

We experiment with 2 different data regimes to get a more complete picture of how each approach scales with available data. In the *high data* regime, we use one update per real environment step and train for a total of 1 million steps. In the *low data* regime, we use 10 updates per step and train for 100 thousand steps. Note that the total number of updates is the same in each case.

**Experiment Design:** For each combination of agent, environment, and data regime we performed an extensive grid search over the Q-network step-size and softmax exploration temperature. We judged these hyperparameters to be the most likely to impact the relative performance of different methods. This grid search was performed for an intermediately complex version of each environment (size 4 ProcMaze, 4 button ButtonGrid, and 7 pipe PanFlute) and the same hyperparameters used for the other complexity levels. We evaluated each hyperparameter setting based on mean final performance of the greedy policy over 30 random seeds. We were able to run 30 seeds efficiently in parallel on a single GPU using automatic batching in JAX (Bradbury et al., 2018). Other hyperparameters, including model step-size, were fixed to reasonable defaults (see Appendix F), not tuned for any specific approach. In Appendix G, we report hyperparameter sensitivity results from this grid search.

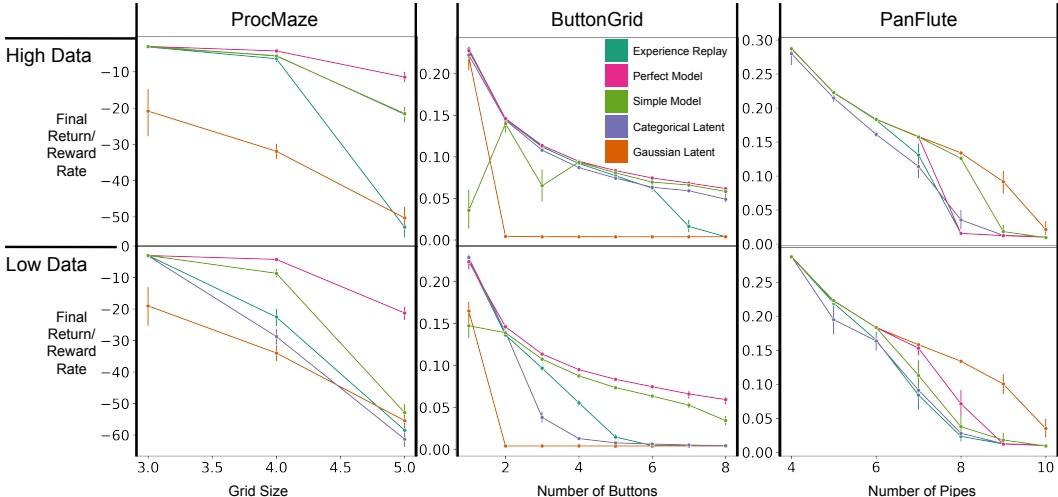

Figure 3: Final performance of greedy policy for all environments in two different data regimes. High data means 1 million environment interactions with 1 update per step, low data means 100 thousand interactions with 10 updates per step. Error bars show 95% confidence intervals.

**Results:** We present results for all environments in Figure 3 (see Appendix H for learning curves). As the results for PanFlute differ substantially from the results for ButtonGrid and ProcMaze, we will discuss them separately. For ButtonGrid and ProcMaze in the high data regime, the simple model and categorical latent model both significantly outperform ER for sufficiently complex environment instances. The Gaussian latent model generally performs quite poorly, which corroborates the results of Hafner et al. (2020) that categorical latents tend to work better in the discrete control setting. Oddly, the simple model also performs much worse for 1 and 3 buttons in ButtonGrid in the high data regime. This may be because less buttons means less examples where a button occupies each particular cell, which makes it harder for the simple model to learn the underlying dynamics.

In the low data regime, the simple model outperforms ER to a greater extent, while the performance of the categorical latent model degrades significantly. This can perhaps be understood by noting that the hypothesis class of the simple model is simpler (the latent-space model can model correlated features, while the simple model cannot) and thus able to generalize well from less data when the simple class is sufficient. Performance of the simple model for 1 and 3 buttons improves when moving to the low data regime. Likely, this indicates underfitting in the high data regime which is helped by training more on each example. Overall, the results for the simple model and categorical latent model in the high data regime, and simple model in the low data regime, show a clear indication of the sample efficiency benefit that can be obtained by using a learned model in these environments.

Results for PanFlute are qualitatively different. Most surprisingly, the Gaussian-latent model and the simple model outperform the perfect model in some of the harder problem instances. This is intuitively strange, and seems to indicate that model errors somehow improve performance.

**Why do Some Learned Models Outperform the Perfect Model on PanFlute?** We hypothesize that the smoother reward and dynamics learned by the model serves as a powerful exploration heuristic. The true reward function is nonzero, and thus the agent receives a learning signal, only in the rare event that every pipe-end is active. The model might instead learn a smoothed reward function, proportional to the number of pipe-ends activated. With this smoothed reward the agent could incrementally learn to activate more pipe-ends, gradually improving toward the correct sequence.

To test this hypothesis, we looked at the models learned at 10,000 time-steps, in 9-pipe PanFlute (high data). We generated a large amount of random trajectory data with a policy that selects actions in alphabetical order with 80% probability and uniformly randomly otherwise to get a good mix of different numbers of active pipe-ends. We bin this data by the number of active pipe-ends and then look at the average model-predicted reward for each bin. Note that the ground truth reward is zero for all expect the 9 active pipe-end bin. To test for favorable smoothing in the transition dynamics, we used the same data. This time, we bin the data by the number of pipe-ends active at the *next* step and look at probability under the model that all pipe-ends are active at the next time-step.

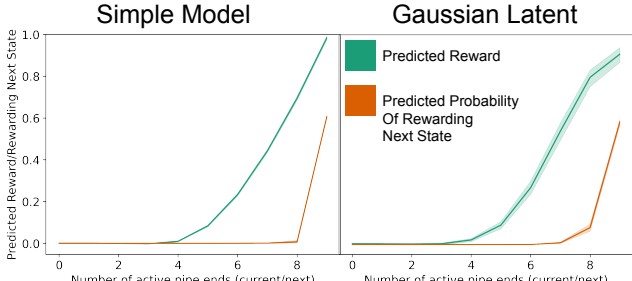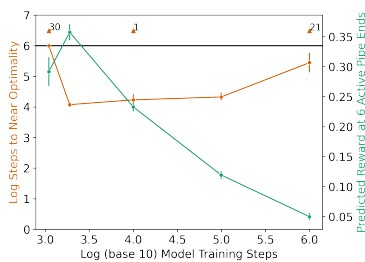

Figure 4: **Left**: Prediction, under simple model and Gaussian latent models, of reward and probability that all pipe-ends are active at the next time step as a function of true number of current and next pipe-ends active respectively on 9-pipe PanFlute. **Right**: In orange, the number of steps to reach near-optimal performance (95% of maximum possible reward rate) on 9-pipe PanFlute when using a frozen simple model trained for a variable number of steps. Numbered arrows indicate the number of seeds out of 30 which failed to reach near-optimal performance within 1 million training steps (marked as black line). In green, the predicted reward for 6 active pipe-ends, displayed as a surrogate for the amount of model smoothing. Error bars show 95% confidence intervals.

The results, for both predicted reward and predicted probability that all pipe-ends will be active at the next time-step ("predicted probability of rewarding next state" in the figure) are displayed in Figure 4(left). We observe that the models indeed tend to learn smoother rewards than the ground truth in a way that might provide a useful exploration heuristic. The Gaussian latent model additionally learns smoothed transition dynamics, predicting all pipe-ends will activate with appreciable probability when in reality only most will, this may provide additional benefit in this problem.

As an additional test of the benefit of model errors, we used simple models frozen at various points in training to train a value function from scratch for 1,000,000 time-steps. We plot when near-optimal performance of the greedy policy is first reached. The results, shown in Figure 4(right) clearly show that a model trained for an intermediate amount of time is most useful for reaching good performance quickly. We also plot the mean predicted reward for states with 6 active pipe-ends for each model as an indication of smoothing, as expected, this decreases with more model training.

The smoothing effect highlighted in these results is interesting for two reasons. First, it may lead model-based algorithms to perform better than expected, acting as a confounding variable when interpreting results. Second, it may be genuinely useful. One could even design algorithms which learn policies within relaxed versions of a model, as a way to drive exploration. However, as there are surely other situations where model smoothing is harmful, this would need to be done with care.

## 5 RELATED WORK

There is a large body of empirical research showing the potential of learned models to improve sample efficiency (Deisenroth & Rasmussen, 2011; Buckman et al., 2018; Kaiser et al., 2019; Janner et al., 2019; Curi et al., 2020; Hafner et al., 2020). A relatively small body of work directly compares ER with learned models. Van Seijen & Sutton (2015) show an exact functional equivalence between a variant of replay with linear TD(0), and linear Dyna (Sutton, 1990). Pan et al. (2018) provide an empirical study comparing ER with learned models under a variety of search control strategies, i.e. different methods for choosing which state-action pairs to update. Holland et al. (2018) empirically compare ER with a parameteric model in the arcade learning environment (ALE; Bellemare et al. (2013)) and highlight the benefits of the model in particular when multi-step rollouts are used. Relative to these works, we focus on understanding *how* a learned model provides a benefit, and highlighting properties of environments where this benefit is most prominent.

Van Hasselt et al. (2019) make a strong case that ER provides many of the benefits of a learned parametric model, and argue that if a model is used only to generate experience starting from observed states it is unlikely to provide additional benefit. We investigate the comparison between ER and learned models further, and argue that there is good reason to believe a learned model can improve sample efficiency over ER in environments with structure, such as factored dynamics. This is true even if the model is used only to augment training data with rollouts starting from observed states.

Dong et al. (2020) share our focus on highlighting situations where model-based RL provides a significant benefit. While we focus on the generalization benefits, they motivate the expressivity benefit of model-based RL in continuous state spaces by showing there exist MDPs where the optimal policy is exponentially more complex to represent than the dynamics.

The benefit of model smoothing, observed in PanFlute, may shed light on some observations in the literature. Hafner et al. (2020) suggest model smoothing as an explanation for why DreamerV2 can achieve good performance on Montezuma's Revenge without any sophisticated exploration mechanism, as is usually required. Holland et al. (2018) observe that their learned model sometimes outperforms the ground-truth model in Seaquest, though they do not suggest a specific explanation.

Our work has implications for *implicit* model-based algorithms (as defined in the survey paper of Moerland et al. (2020)) such as MuZero (Schrittwieser et al., 2020). In such algorithms, the model is not trained to predict future observations, but only task relevant aspects of the future such as policy, value and reward. In light of Theorem 1, and the example in Section 2, it is unclear whether such techniques fully exploit the benefits of model-based learning due to the limited training signal used to constrain the model.[4] Indeed, there is empirical evidence suggesting that MuZero's sample efficiency can be improved by using additional training signals (Ye et al., 2021; Anand et al., 2022).

Our work is loosely inspired by the literature on exploiting factored structure in MDPs (Sallans & Hinton, 2004; Strehl et al., 2007; Diuk et al., 2009; Osband & Van Roy, 2014; Sun et al., 2019; Xu & Tewari, 2020). Theorem 1, in particular, is loosely related to Theorem 2 of Sun et al. (2019), which shows that there exists a family of MDPs where a model-based approach can be exponentially more efficient than any model-free approach. We focus on factored structure to motivate the benefit of model generalization, but do not use algorithms explicitly designed to exploit this factored structure.

AIXI (Hutter, 2004) provides a particularly general approach to model-based RL. An AIXI agent maximizes expected return over all computable world models with prior probability given by Solomonoff's universal prior (Solomonoff, 1978), which assigns higher probability to models with lower Kolmogorov complexity. While AIXI is itself not computable, computable approximations exist (Veness et al., 2011). Here, we take a step back from considering all computable models, and highlight the benefit of model learning in general, relative to model-free learning with ER. We focus on understanding how model-based learning can facilitate generalization, and thus improve sample efficiency, in a way that model-free learning does not. Our empirical results suggest this applies even for the practical subset of computable models representable by a simple NN architecture.

In this work, we focus on the advantages of using models to generate experience for policy improvement. Learned models can also be useful in other ways, for example, local planning for immediate action selection (Richalet et al., 1978; Chua et al., 2018; Byravan et al., 2021), improving credit assignment (Schmidhuber, 1990; Heess et al., 2015; Buesing et al., 2018), exploration (Schmidhuber, 1990; 1997; Pathak et al., 2017), representation learning (Lin & Mitchell, 1992; Gregor et al., 2019; Hessel et al., 2021), and in answering learned queries (Schmidhuber, 2015).

## 6 CONCLUSION

In Theorem 1 we highlighted how, for a model class with some known structure, we can narrow down the possible value functions more with a model-based approach than a model-free approach. We provided empirical evidence that the intuition behind this theorem holds in practice when we use NN function approximation in domains with factored structure. In such cases, we have verified through extensive experiments that a model-based method can maintain strong performance as the complexity of the environment increases beyond the point where an analogous model-free approach fails. As an aside we demonstrated how, by smoothing the reward and/or transition dynamics, experience generated by a learned model can provide a useful signal for exploration that can sometimes lead to better performance than even a perfect model. Overall, we believe that our work can help to ground future work in model-based RL in a better understanding of how learned models can improve sample efficiency. An interesting direction for future work lies in better understanding the inductive biases that allow simple NN models to generalize in a way that allows them to efficiently learn factored structure, and how more sophisticated architectures could improve on this.

---

[4]Though Gehring et al. (2021) suggest that the implicit model-based parameterization itself may yield favorable gradient dynamics.

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

# A  THEOREM MOTIVATING THE BENEFIT OF MODEL GENERALIZATION

Here, we present and prove a simple theorem motivating the benefit of learning a parametric model over learning a value function directly from ER. Intuitively speaking, the theorem states that, when narrowing down the set of possible value functions based on observed data, we can rule out more if we first rule out models directly, and demand the value function be consistent with the reduced model class, than if we only demand the value function obeys the Bellman optimality equation with respect to observed transitions.

We state the theorem within the formalism of finite MDPs. An MDP consists of a state-space $\mathcal{S}$, action-space $\mathcal{A}$, reward function $r : \mathcal{S} \times \mathcal{A} \to \mathbb{R}$ and transition function $p$. In general, $p$ maps state-actions pairs to probability distributions over possible next states. However, in Theorem 1, we consider deterministic MDPs, meaning each state-action pair maps to a distribution with probability one on a particular next state $s'$ and zero for all other next states. In this case, it will be convenient to write $p : \mathcal{S} \times \mathcal{A} \to \mathcal{S}$ as a mapping from state-action pairs to the only possible next state. We assume $\mathcal{S}$ and $\mathcal{A}$ are finite sets for simplicity.

An agent interacts with an MDP in a series of time-steps, beginning in some state $S_0 \in \mathcal{S}$. At each time-step, the agent observes the current state $S_t \in \mathcal{S}$ and selects an action $A_t \in \mathcal{A}$ in response. The environment then transitions to the next state $S_{t+1} = p(S_t, A_t)$ and the agent receives a reward $R_{t+1} = r(S_t, A_t)$. Interaction continues until some designated terminal state $S_T = \perp$ is reached at which point the episode is over. The goal of an agent is to find a policy, a mapping from states to actions[5], $\pi : s \to a$ which maximizes the expected return $\mathbb{E}_\pi[G_t|S_t = s]$, where $G_t = \sum_{k=t+1}^{T} R_k$, from any given state until the terminal state $S_T = \perp$ is reached when actions are selected according to $\pi$. We assume that all policies eventually reach $\perp$ with probability one, such that the return is well defined. The action-value function of a policy is defined as $q_\pi(s, a) = \mathbb{E}_\pi[G_t|S_t = s, A_t = a]$, the expected return if action a is selected in state s and policy $\pi$ is followed from that point forward.

The optimal action-value function $q^\star(s, a) = \max_\pi q_\pi(s, a)$ is defined as the maximum action value over all policies $\pi$. $q^\star(s, a)$ is known to be the unique solution to the Bellman optimality equation $q^\star(s, a) = r(s, a) + \mathbb{E}[\max_{a'} q^\star(S_{t+1}, a')|S_t = s, A_t = a]$ where the value of all actions in the terminal state $\perp$ are defined to be zero. In the case of a deterministic transition function, this reduces to simply $q^\star(s, a) = r(s, a) + \max_{a'} q^\star(p(s, a), a')$.

We state Theorem 1 for deterministic MDPs, but analogous results likely hold for general MDPs, albeit significantly complicated by the fact that in the general case, models and value functions can only be ruled out with high probability based on observed data, as opposed to with certainty.

**Theorem 1.** *Consider a class of episodic MDPs, $\mathcal{M}$, with fixed reward function $r : \mathcal{S} \times \mathcal{A} \to \mathbb{R}$ , and deterministic transition function belonging to a hypothesis class $H \subseteq \{p : \mathcal{S} \times \mathcal{A} \to S\}$. Assume, for all $p \in H$, all policies lead to eventual termination.*

*Since each MDP in the class is deterministic, following a deterministic policy $\pi$, beginning in $s, a$ will give rise to a specific state-action trajectory $\tau_p(\pi, s, a) \doteq (s_0, a_0, s_1, a_1, ... s_{T-1}, a_{T-1}, \perp)$ where $s_0 = s, a_0 = a$, $\perp$ is the terminal state, and $p \in H$. Define also $G(\tau_p(\pi, s, a)) \doteq \sum_{t=0}^{T-1} r(s_t, a_t)$, the return associated with the trajectory.*

*Next, define the class of optimal action-value functions associated with $H$:*

$$H_Q \doteq \{q : s, a \to \mathbb{R} | \exists p \in H : \forall s, a \; q(s, a) = \max_\pi G(\tau_p(\pi, s, a))\},$$

*or equivalently:*

$$H_Q = \{q : s, a \to \mathbb{R} | \exists p \in H : \forall s, a \; q(s, a) = r(s, a) + \max_{a'} q(p(s, a), a')\}.$$

*Consider a dataset $D = \{(s_n, a_n, s'_n) | n \in \{0, 1, .., N\}\}$ of transitions such that $p(s_n, a_n) = s'_n$ for some $p \in H$. We now define two different notions of hypothesis classes over action-value functions which are consistent with $D$:*

$$H_B(D) = \{q \in H_Q | \forall n \; q(s_n, a_n) = r(s_n, a_n) + \max_{a'} q(s'_n, a')\}$$

$$H_M(D) = \{q \in H_Q | \exists p \in H : (\forall n \; p(s_n, a_n) = s'_n) \wedge (\forall s, a \; q(s, a) = r(s, a) + \max_{a'} q(p(s, a), a'))\}$$

---

[5]Again, policies can more generally map states to distributions over actions, but we focus here on the deterministic case, and note that there is always a deterministic optimal policy.

*Where $B$ stands for Bellman consistency and $M$ stands for model consistency. In words, these are the hypothesis classes consisting of value functions which obey the Bellman optimality equation with respect to the observed transitions, and the hypothesis class consisting of true optimal value functions for transition dynamics which are consistent with the observed transitions respectively. Then the following are true:*

1. *$H_M(D) \subseteq H_B(D)$.*
2. *For some choices of $\mathcal{M}$ and $D$, $H_M(D) \subset H_B(D)$.*
3. *For a tabular transition function class, that is one that includes every possible mapping from state-action pairs to next states, $H_M(D) = H_B(D)$.*

*Proof.* We begin by proving **part 1**. Towards this, assume $q \in H_M(D)$. Then by definition we have the following:

$$\exists p \in H : (\forall n \; p(s_n, a_n) = s'_n) \wedge (\forall s, a \; q(s, a) = r(s, a) + \max_{a'} q(p(s, a), a'))$$

$$\implies \exists p \in H : (\forall n \; p(s_n, a_n) = s'_n) \wedge (\forall n \; q(s_n, a_n) = r(s_n, a_n) + \max_{a'} q(p(s_n, a_n), a'))$$

$$\implies \forall n \; q(s_n, a_n) = r(s_n, a_n) + \max_{a'} q(s'_n, a')$$

$$\implies q \in H_B(D),$$

which proves part 1.

To prove **part 2**, it suffices to construct a specific $\mathcal{M}$ and $D$ for which $H_M(D) \subset H_B(D)$. Towards this, consider a class of deterministic MDPs $\mathcal{M}$ with state-space $\mathcal{S} \subset \mathbb{N}^3 \cup \perp$ where the dynamics of each component of (nonterminal) $S_t \in \mathcal{S}$ are factored such that $S_t[i]$ does not influence $S_{t+1}[j]$ for $i \neq j$. Furthermore $S_t[0] \in \{0, 1, 2\}$, $S_t[1] \in \{0, 1\}$, $S_t[2] \in \{0, 1\}$. The reward function is defined as follows:

$$r(s, a) = \begin{cases} 1 \text{ if } s[1] = s[2] = 1 \text{ and } s[0] = 0 \\ 0 \text{ otherwise.} \end{cases}$$

The transition dynamics for $S_t[0]$ are fixed within the class such that

$$S_{t+1}[0] = S_t[0] - 1 \text{ if } S_t[0] > 0$$
$$S_{t+1} = \perp \text{ otherwise.}$$

Thus $S_t[0]$ acts as a counter for the number of steps until termination, and the agent gets a positive reward only if both bits $S_{T-1}[1]$ and $S_{T-1}[2]$ are set to 1 at the step prior to termination. The action space is $\mathcal{A} = \{0, 1, 2\}$. This model class is illustrated in Figure 5.

Now let the dataset D consist of the following transitions:

$$(s : (2, 0, 0), a : 0, s' : (1, 1, 0))$$
$$(s : (2, 1, 1), a : 0, s' : (1, 0, 1))$$
$$(s : (2, 0, 0), a : 1, s' : (1, 0, 1))$$
$$(s : (2, 1, 1), a : 1, s' : (1, 1, 0))$$
$$(s : (2, 0, 0), a : 2, s' : (1, 0, 0))$$
$$(s : (2, 1, 1), a : 2, s' : (1, 1, 1)). \tag{1}$$

Given the factored dynamics and that the dynamics of $S_t[0]$ are fixed within the model class, this data suffices to uniquely determine $p \in H$ to be the transition function identified by the following equations (when $S_{t+1}$ is nonterminal), in addition to the known dynamics for $S_t[0]$:

$$S_{t+1}[1] = \begin{cases} \text{not}(S_t[1]) \text{ if } A_t = 0 \\ S_t[1] \text{ otherwise} \end{cases}$$

$$S_{t+1}[2] = \begin{cases} \text{not}(S_t[2]) \text{ if } A_t = 1 \\ S_t[2] \text{ otherwise.} \end{cases}$$

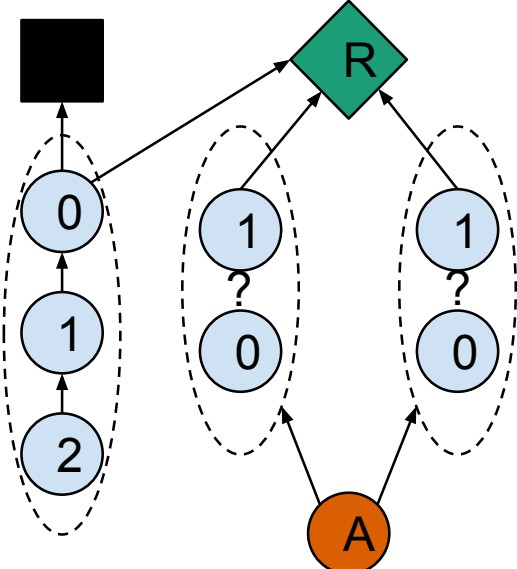

Figure 5: Illustration of the model class used as an example where selecting hypothesis based on model consistency is more selective than Bellman consistency. The state consists of three components from left to right. The left most component is fixed within the class and simply acts as a countdown to termination. The other two components have unknown internal transition dynamics but are known to not be influenced by other state components. The reward function is known to be zero except for when the two rightmost components are set to one at termination.

That is $A_t = 0$ toggles component 1 of the state and $A_t = 1$ toggles component 2, while $A_t = 2$ leaves both components unchanged. In particular, the data is such that it contains an example of the effect of each action on each possible bit value of $S_t[1]$ and $S_t[2]$, meaning the above transition function is the only possibility. Thus $H_M(D)$ consists of the singleton of the true optimal value function $q^*(s, a)$ for this MDP, which is 1 if and only if the agent has sufficient steps remaining after taking action $a$ before termination to toggle bits $S_{T-1}[1]$ and $S_{T-1}[2]$ both to 1.

On the other hand, we can show that $H_B(D)$ is a larger set. In particular, consider the following action-value function:

$$\hat{q}(s, a) = \begin{cases} 1 \text{ if } s[1] = s[2] = 1 \text{ and } s[0] = 0 \\ 0 \text{ otherwise.} \end{cases} \tag{2}$$

Note that $\hat{q}(s, a) \in H_B(D)$, as it satisfies the Bellman equation,

$$\hat{q}(s_n, a_n) = r(s_n, a_n) + \max_{a'} \hat{q}(s'_n, a'),$$

for all transitions in $D$, although it is not the true optimal value function and thus is not in $H_M(D)$. The value function $\hat{q}(s, a)$ is also in $H_Q$, since it is the value function of the MDP with alternative factored transition dynamics where (for example) $S_{t+1}[1] = S_{t+1}[2] = 0$ regardless of their previous value or the action choice. This suffices to prove part 2 of the Theorem.

Finally, we prove **part 3**. Since we have already shown in part 1 that $q \in H_M(D) \implies q \in H_B(D)$ in the general case, it suffices to show that, with the additional restriction of a tabular class $H$ of transition matrices, we have $q \in H_B(D) \implies q \in H_M(D)$. Recall that by a tabular $H$ we mean one which includes every possible mapping from state-action pairs to next states. Thus, in this case we are free to choose $p(s, a)$ to be an independently selected $s'$ for every $(s, a)$ pair, and know that the resulting $p \in H$. Assume $q \in H_B(D)$, then by definition we know $q \in H_Q$, meaning

$$\exists p \in H : \forall s, a \, q(s, a) = r(s, a) + \max_{a'} q(p(s, a), a'), \tag{3}$$

and, by the definition of $q \in H_B(D)$, we know

$$\forall n \, q(s_n, a_n) = r(s_n, a_n) + \max_{a'} q(s'_n, a'). \tag{4}$$

Now, given tabular $H$ we can set, for all $n$, $p(s_n, a_n) = s'_n$, which we know from Equation 4 gives us:

$$q(s_n, a_n) = r(s_n, a_n) + \max_{a'} q(p(s_n, a_n), a').$$

Now for $s, a$ where $\nexists n : (s, a) = (s_n, a_n)$, given Equation 3, we know that for some choice of $p(s, a)$ it holds that:

$$q(s, a) = r(s, a) + \max_{a'} q(p(s, a), a').$$

Thus, given the choice of tabular $H$, we can choose $p$ to simultaneously satisfy $p(s_n, a_n) = s'_n$ for all $n$ and the Bellman optimality equation for all $(s, a)$, and indeed:

$$
\begin{aligned}
&q \in H_B(D) \\
&\implies \exists p \in H : (\forall n \; p(s_n, a_n) = s'_n) \wedge (\forall s, a \; q(s, a) = r(s, a) + \max_{a'} q(p(s, a), a')) \\
&\implies q \in H_M(D),
\end{aligned}
$$

which completes the proof of part 3. $\qquad\square$

Note that, for simplicity, the dataset $D$ in the example for part 2 does not consist of full episodic trajectories. However, it is straightforward to come up with a dataset of episodic trajectories for which the same outcome still holds, for example by appending each transition in Equation 1 with any sequence of transitions leading to termination with $0$ reward.

Avoiding inclusion of the rewarding transition in the dataset is also not necessary to obtain an analogous result. For example, consider adding another action $a = 3$ to the above example with known dynamics that always switch $S_{t+1}[1]$ and $S_{t+1}[2]$ to 1, but always gives an immediate reward of $-1$. Now consider adding the following episodic sequence (here including rewards for clarity) to the dataset of Equation 1:

$$
\begin{aligned}
&(s : (2, 0, 0), a : 2, r : 0, s' : (1, 0, 0)) \\
&(s : (1, 0, 0), a : 3, r : -1, s' : (0, 1, 1)) \\
&(s : (0, 1, 1), a : 2, r : 1, s' : \perp).
\end{aligned}
$$

Though the data still uniquely specifies the model, $H_B(D)$ will contain the incorrect action-value function

$$
\hat{q}(s, a) = \begin{cases}
-1 & \text{if } a = 3 \text{ and } s[0] \neq 1 \\
1 & \text{if } s[1] = s[2] = 1 \text{ and } s[0] = 0 \text{ and } a \neq 3 \\
0 & \text{otherwise,}
\end{cases}
$$

which is a minor modification of Equation 2 to include the case where $a = 3$.

It is also straightforward to come up with similar examples where the data uniquely specifies a model, but $H_B(D)$ includes value functions for which the associated greedy policy is unique and suboptimal in the true model.

Practical learning algorithms like stochastic gradient descent applied with NN function approximation don't work by directly ruling out hypotheses based on the data. Rather than considering models within an explicitly defined class, a NN would have some inductive bias towards particular models and away from others depending on the architecture and hyperparameters. Nonetheless, the idea behind Theorem 1 helps to give insight into why we should expect learning a parametric model to provide a sample efficiency benefit.

Intuitively, we can think of the process of performing many updates to a sufficiently high capacity neural network, with data from a dataset, as incrementally constraining the possible functions represented by the network. The specific function converged to will depend on the initialization and random batches selected for updates. Theorem 1 suggests that learning a model as an intermediate step can impose more constraints on the possible value functions. This in turn should increase the chance of converging to a value function that generalizes well from limited data.

## B   Further Environment Details

The environments we investigate are all Markov and use flat binary observation vectors. To allow the model-based agent to learn about the reward or termination function, we include occasional random transitions to rewarding or terminal states. The probability of these transitions is chosen to result in much lower return than is achievable with optimal performance in each environment. Except for these random transitions, all environments are deterministic (with the exception of rare random transitions to rewarding or terminal states) so simple models can be expected to work well, though we also investigate more sophisticated latent-space models. Here, we will describe each of the environments in some detail.

The first environment, ProcMaze, is illustrated in Figure 2(left). ProcMaze is an episodic environment. ProcMaze consists of procedurally generated gridworld mazes, where an agent has to navigate from a start state to a goal state. The maze itself, along with the start state and goal state are randomized at the start of each episode. A reward of -1 is given for each step until the goal is reached, at which point the episode terminates. Also the agent is rarely randomly teleported to the goal (probability $0.1/T$ where $T$ is the time required to complete the worst case problem instance for the grid size under the optimal policy) such that it can obtain knowledge of the reward function even with a poor behavior policy. Difficulty could be scaled by increasing the grid size. The observations consist of a flat binary vector including: one hot vectors for the goal location and agent location, a vector which is one if and only if a cell contains a wall, and a vector which is one if and only if a cell does not contain a wall. The action space includes attempting to move in each cardinal direction, and no-op. An attempted move will fail if it would lead the agent into a wall or the edge of the grid. In each episode a new maze is generated using randomized depth first search, which produces reasonable mazes and guarantees the goal is reachable.

The second environment, ButtonGrid, is illustrated in Figure 2(middle). ButtonGrid is a continuing environment, with no termination. ButtonGrid consists of a grid world with a set of randomly placed buttons. An agent (orange in the figure) can move around on the grid and if it hits a button it will toggle it either on (black in the figure) or off (white in the figure). Reward is given whenever all the buttons are set to on, at which point and the button locations are randomized, but the number of buttons held fixed, and all buttons set to off. Occasionally all the buttons will spontaneously switch to on (probability $0.1/(\text{grid size})^2$), meaning the agent can receive examples of the reward function even while behaving suboptimally. Importantly, it does not suffice to touch each button once to solve this environment, as they are toggled on and off by repeated contact, an agent must also carefully avoid hitting them again after the first time they are pressed. Difficulty can be scaled by increasing the number of buttons on the grid, as well as the grid size, but we focus on the former. The observations consist of a flat binary vector including: one hot vectors for the agent location, a vector which is one if and only if a cell contains a button which is turned on, and a vector which is one if and only if a cell contains a button which is turned off. The action space includes attempting to move in each cardinal direction, and no-op. An attempted move will fail only if it would lead the agent into the edge of the grid.

The third environment, PanFlute is illustrated in Figure 2(right). PanFlute is a continuing environment with no termination. PanFlute is intended as a minimal instantiation of an environment with combinatorial complexity in terms of optimal behavior, but a simple factored transition structure. The observations consists of a binary value for each square in the figure. An agent has n actions available to it, (a,b,c,d,e) in the figure. Each action will activate the associated cell at the bottom of a specific *pipe*. The pipe associated with the last action (alphabetically) has a length of one cell, every other pipe is one cell longer than its (alphabetical) successor. If a cell is activated at a given time-step, it will deactivate and activate the cell above it in the same pipe at the next time-step. A reward of 1 is received if the cells at the end of each pipe are simultaneously active, otherwise the reward is always zero. An active pipe-end will always deactivate at the next step regardless of whether reward is obtained. Occasionally, the cells at the end of all pipes will activate spontaneously (probability $1/n^2$), thus allowing the agent to observe a rewarding situation without having to create it through its own actions. Otherwise, due to the arrangement of pipe lengths, the only way for the agent to obtain reward is to choose each of the $n$ actions in sequence. We can scale the difficulty of the environment by changing the number of actions $n$. The probability of a random sequence of $n$ actions reaching the rewarding state (aside from spontaneous activation) is $1/n^n$. Observations consist of a flat binary vector which includes the active/inactive state of each cell in each pipe.

## C  Experiments in an Environment Without Structured Transitions

This work primarily focuses on highlighting environments in which model-based learning is expected to be beneficial. Nevertheless, it is worthwhile to contrast this with what happens in environments which do not have such favorable characteristics. To that end, we ran an additional experiment on an environment without factored structure, which we will now present.

The environment for this experiment, which we refer to as OpenGrid, was simply an open grid with a goal in the bottom right corner and a reward of $-1$ for every step until the goal is reached at which point termination occurs. The agent starts in a random location in each episode. As in our other experiments, we include occasional spontaneous transitions to the goal (probability $0.1/(\text{grid size})$). The agent location is simply represented by a one-hot vector (effectively tabular) so there is really no structure to exploit. The learned model must essentially memorize every individual transition to learn the dynamics.

Our experimental design was the same as in Section 4. We tune the Q-network step size and softmax exploration temperature from the same set of values on a grid size of size 12 and then used the best hyperparameters for each agent on the other grid sizes. In this experiment, we focused on the low-data regime where the simple model tended to have the biggest advantage over ER in our other experiments.

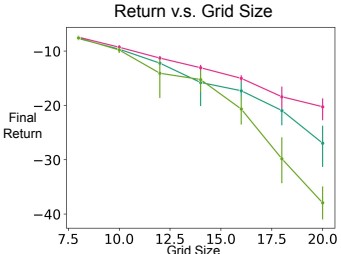 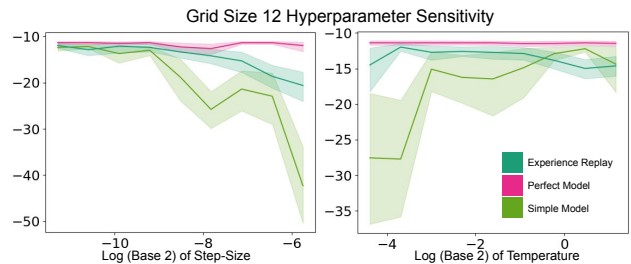

Figure 6: **Left**: Final performance of greedy policy v.s grid size for OpenGrid in the low data regimes, that is 100 thousand interactions with 10 updates per step. **Right**: Softmax temperature and step-size sensitivity curves for each approach resulting from the grid-search on an size 12 OpenGrid. In these plots, the other hyperparameter is fixed to its best value from the grid-search while varying the temperature or step-size. Error bars show 95% confidence intervals.

The results are shown in Figure 6, with performance v.s. grid-size on the left and hyperparameter sensitivity curves from the initial tuning on the right. In contrast to our other experiments, here we see that the simple model becomes worse relative to ER as the environment complexity increases. This is reasonable as the model has no ability to extrapolate beyond the data. The best it can do is memorize what is already in the replay buffer and the limitations of finite model capacity and imperfect optimization prevent it from doing so perfectly. This result helps to contextualize our main results for environments with factored structure by showing how the performance of the model-based approach suffers in a simple environment without such structure.

## D  Model Details

Our latent-space model consists of the following components:

- Representation Model: $\phi_t \sim q_\theta(\phi_t|o_t)$
- Observation Reconstructor: $\hat{o}_t \sim p_\theta(\hat{o}_t|\phi_t)$
- Transition Predictor: $\hat{\phi}_t \sim p_\theta(\hat{\phi}_t|\phi_{t-1}, a_{t-1})$
- Reward Predictor: $\hat{r}_t \sim p_\theta(\hat{r}_t|\phi_{t-1}, a_{t-1})$
- Termination Predictor: $\hat{\gamma}_t \sim p_\theta(\hat{\gamma}_t|\phi_{t-1}, a_{t-1})$.

All components are implemented as NNs with $\theta$ representing the combined parameter vector. $o_t$ is the observation from the environment at time $t$, $a_t$ is the action, $r_t$ is the reward, $\gamma_t$ is the continu-

ation probability.[6] $\phi_t$ is the latent-state constructed by the model from the observation from which transitions, rewards and terminations are all predicted. The associated versions of each variable with hats are predictions made by the model. The Q-network associated with the latent-space models are always trained to predict action-values directly from $\phi_t$ as opposed to first reconstructing the observation.

For training the model, we use a loss very similar to that employed by Hafner et al. (2020):

$$\mathcal{L}_t(\theta) = -\log(p_\theta(o_t|\phi_t)) - \log(p_\theta(r_t|\phi_{t-1}, a_{t-1})) - \log(p_\theta(\gamma_t|\phi_{t-1}, a_{t-1})) \\ + KL(q_\theta(\phi_t|o_t)|p_\theta(\phi_t|\phi_{t-1}, a_{t-1})),$$

where $\phi_t \sim q_\theta(\phi_t|o_t)$. We also employ KL-balancing, as described by Hafner et al. (2020), with $\alpha = 0.8$. We experiment with both Gaussian and Categorical latent variables for $\phi_t$. For the categorical case we use the straight-through estimator to propagate gradients through the discrete latent variables where necessary. In all cases, we train the model on randomly sampled transition from a replay-buffer. Note that it is not necessary to train on sequences in our case, as the lack of recurrence means that the loss at each time-step can be independently evaluated. The observation reconstructor $p_\theta(\hat{o}_t|\phi_t)$ uses a sigmoid activation to output the means of a vector of Bernoulli distributions since all tested environments use binary observations. The reward predictor $p_\theta(\hat{r}_t|\phi_{t-1}, a_{t-1})$ uses a linear activation and outputs the mean of a univariate Gaussian, in which case the above loss is effectively mean-squared error. The termination predictor $p_\theta(\hat{\gamma}_t|\phi_{t-1}, a_{t-1})$ uses a sigmoid activation to output a single Bernoulli termination probability.

For the simple model, the reward and termination predictors are the same except that they take raw observations as input instead of latent variables. Likewise the transition predictor $p_\theta(\hat{o}_t|o_{t-1}, a_{t-1})$ works directly in observation space, and is trained with a negative log-likelihood loss relative to the true observations.

## E ILLUSTRATIVE EXPERIMENT DETAILS

Here we give some additional detail on the setup of the illustrative experiment in Section 2. For the most part, we used the same hyperparameters as our main experiments, as detailed in Appendix F. The only exception is that for this simple experiment we did not tune any hyperparameters, but rather fixed the Q-learning step-size to $2e - 4$ and softmax exploration temperature to $0.1$. The softmax exploration temperature only applies to the model in this case, since the model-free agent was trained on fixed data and thus never selects actions during learning.

We train all agents for 1,000,000 training steps, to insure convergence, which was excessive given the small fixed dataset used for training. To control for the total number of value function updates, and the total number of (real or imagined) transitions used in each update, we made the following choices: for model-free DQN and the 1-step model-based agent the value function is update on a batches of 320 transitions, from the dataset or the learned model; the 10-step model-based agent uses a batch of 32 sequences of length 10 generated by the model to equate the number of transitions per update. The models are always trained on 32 transitions from the dataset in each update.

---

[6]Two out of three of the environments we experiment with are continuing, thus termination will never occur and this prediction could be omitted, however, $\gamma_t = 1$ should be learned easily and thus it should make little difference whether it is included or not.

## F    HYPERPARAMETERS

| Shared Hyperparameters | Value |
|---|---|
| Number of Hidden Layers | 3 |
| Number of Hidden Units | 200 |
| Hidden Activation | ELU |
| Optimizer | AdamW |
| Adam $\beta_1$ | 0.9 |
| Adam $\beta_2$ | 0.99 |
| Adam $\epsilon$ | 1e-5 |
| Adam weight decay | 1e-6 |
| Q-learning Step-Size | **Tuned** (See Appendix G) |
| Discount Factor | 0.9 |
| Batch Size | 32 for model-based, 320 for model-free |
| Exploration Strategy | Softmax |
| Softmax Temperature | **Tuned** (See Appendix G) |
| Target Network Update Frequency | 100 |
| Buffer Size | 100,000 |
| Training Start Time | 1000 |
| **Model Hyperparameters** | |
| Model Learning Step-Size | 2e-4 |
| Rollout Length | 10 |
| **Categorical Latent Hyperparameters** | |
| Number of Features | 32 |
| Width of Features | 32 |
| KL Balancing | 0.8 |
| KL Loss Scale | 1.0 |
| **Gaussian Latent Hyperparameters** | |
| Number of Features | 32 |
| KL Balancing | 0.8 |
| KL Loss Scale | 1.0 |
| Minimum std | 0.1 |
| std activation | $2 \cdot \sigma(x/2)$ |

Table 1: Table of hyperparameters used in experiments in Section 4.

## G    HYPERPARAMETER TUNING AND SENSITIVITY EXPERIMENTS

Here, we present hyperparameter sensitivity plots for step-size and softmax exploration temperature which resulted from our initial grid-search to select hyperparameters for the main experiments in Section 4. The initial grid-search tried the set $\{0.0125, 0.025, 0.05, 0.1, 0.2, 0.4, 0.8, 1.6, 3.2\}$ for the softmax temperature and the set $\{1.25e-05, 2.50e-05, 5.00e-05, 1.00e-04, 2.00e-04, 4.00e-04, 8.00e-04, 1.60e-03, 3.20e-03\}$ for the step-size. This range was extended in some cases where there was a significant positive trend at the boundary of the range for either parameter, this only effected results for ER and the Gaussian latent model and the impact was negligible compared to using the best parameters in the initial range. In each case, the mean performance of 30 random seeds was used to evaluate each hyperparameter setting in terms of final performances of the greedy policy. The hyperparameters with the best final performance were selected in each case for use in our main experiments. Sensitivity curves for step-size and temperature are shown in Figure 7 and Figure 8 respectively.

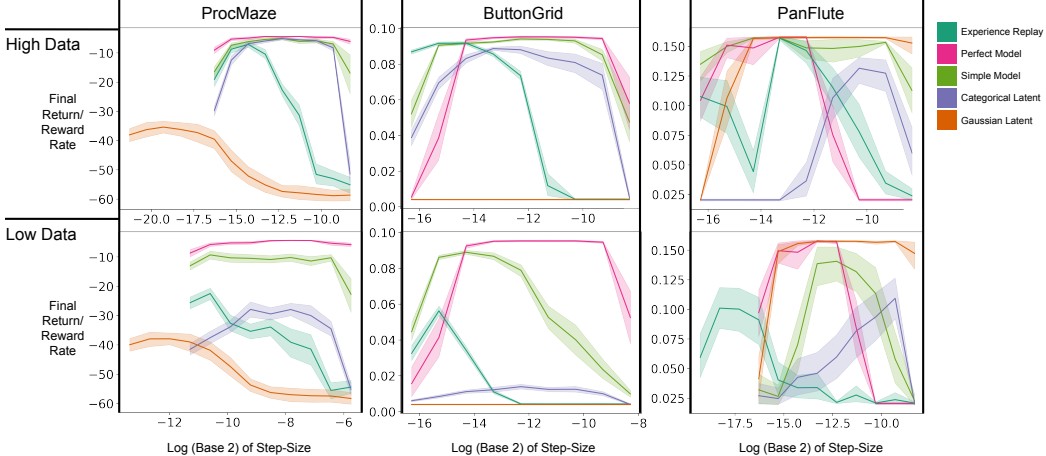

Figure 7: Step-size sensitivity curves for each approach resulting from the grid-search on an intermediate level of difficulty for each environment (size 4 ProcMaze, 4 button ButtonGrid, 7 pipe PanFlute). In these plots, the softmax temperature is fixed to its best value from the grid-search while varying step-size. The search was extended in a few cases when there was a significant positive trend at the boundary of the initial search grid. Error bars show 95% confidence intervals.

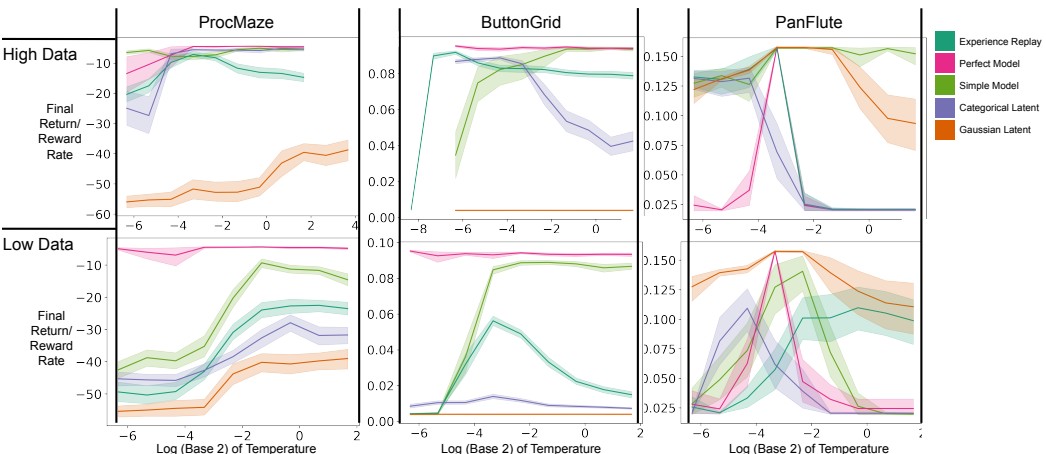

Figure 8: Softmax temperature sensitivity curves for each approach resulting from the grid-search on an intermediate level of difficulty for each environment (size 4 ProcMaze, 4 button ButtonGrid, 7 pipe PanFlute). In these plots, the step-size is fixed to its best value from the grid-search while varying softmax termperature. The search was extended in a few cases when there was a significant positive trend at the boundary of the search grid. Error bars show 95% confidence intervals.

## H LEARNING CURVES

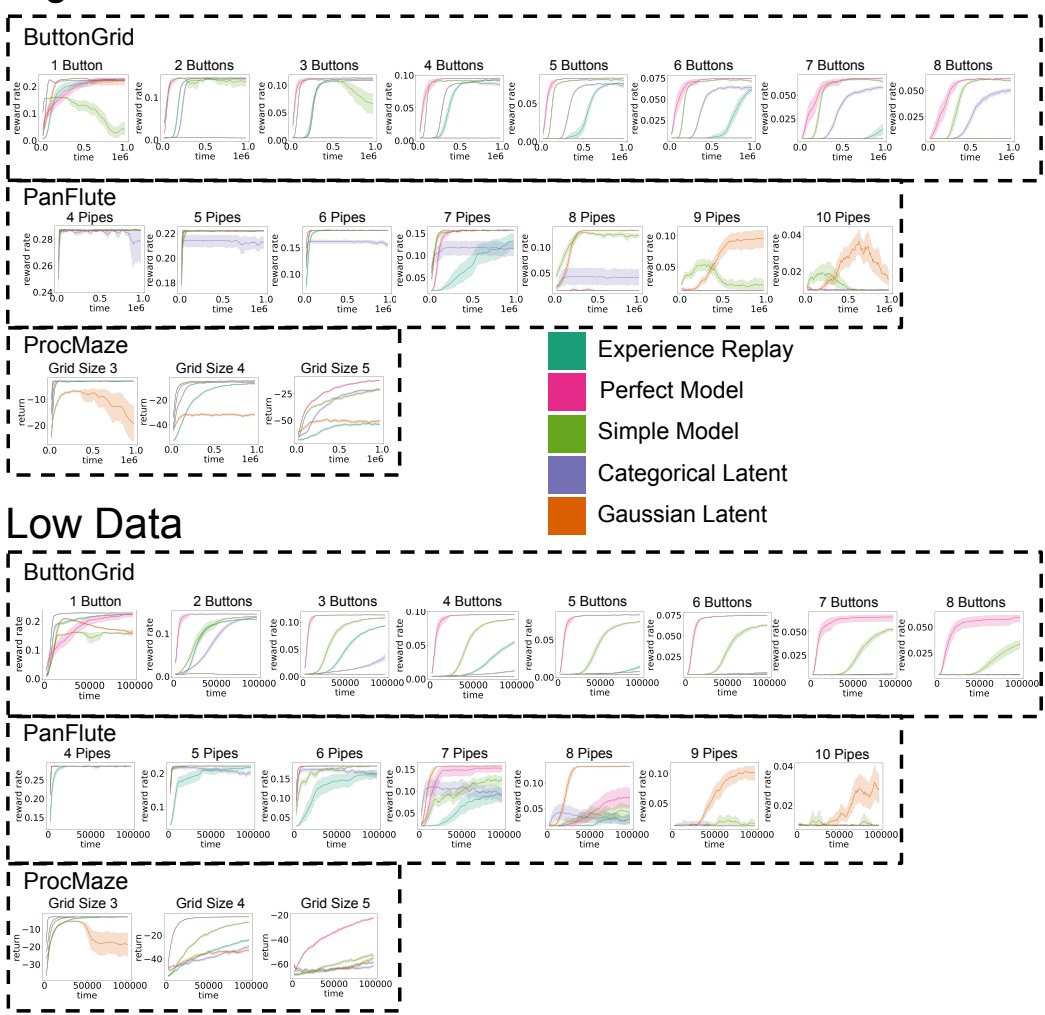

Figure 9: Full learning curves for experiments in Section 4. Performance of the greedy policy is plotted every 5000 updates and smoothed with a moving average over the last 10 values. Error bars show 95% confidence intervals.

## I ABLATION EXPERIMENTS

In this section, we present some additional ablation studies to better understand the impact of some of our experimental design decisions made in the main paper. Figure 10 highlights the impact of removing spontaneous transitions to rewarding states in each of the environments. Figure 11 compares the performance of the simple model with 1-step and 10-step model rollouts, where the total number of simulated transitions used in each batch used for DQN updates is controlled.

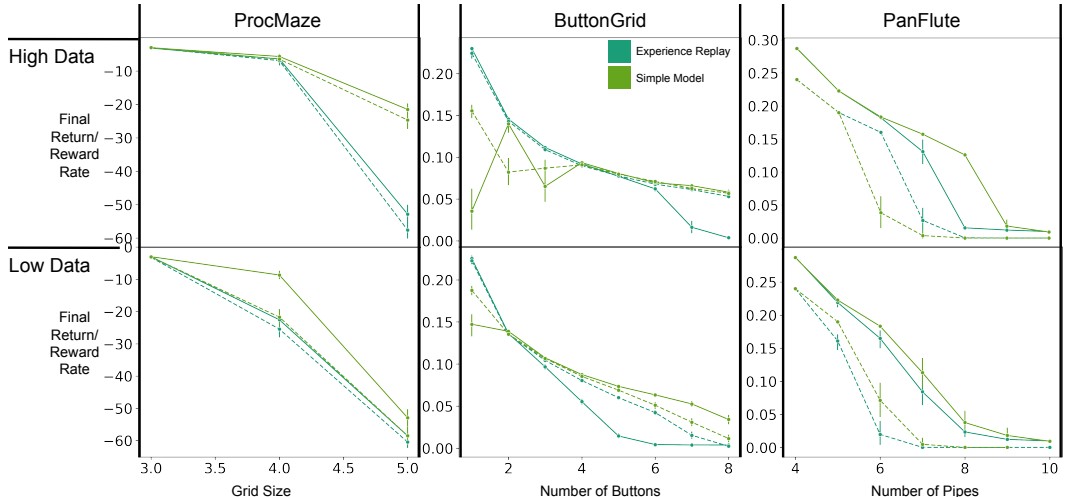

Figure 10: Comparing the performance of ER and the simple model in variants of each environment class with spontaneous rewards disabled to the original environment. Dotted lines show the curve with spontaneous rewards disabled. The effect of this is highly variable, but is generally detrimental to the model-based approach. Perhaps surprisingly, ER appears to perform better without the random transitions to rewarding states in ButtonGrid, perhaps due to elimination of the resulting noise from the learning signal. The impact was largest in PanFlute, where the absence of spontaneous rewards negatively effected both model-free and model-based approaches, but in the high-data regime lead model-free to outperform model-based. Error bars show 95% confidence intervals.

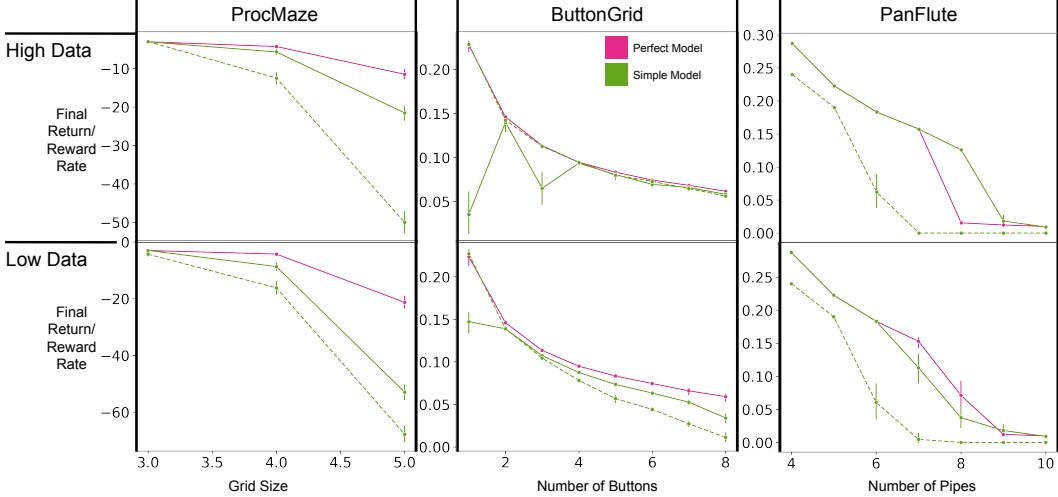

Figure 11: Comparing simple model performance with 1-step and 10-step rollouts, with perfect model included for reference. Dotted line is 1-step rollouts, solid line is 10-step. In ProcMaze and PanFlute, 10-step rollouts consistently perform much better, however is ButtonGrid 1-step rollouts perform better for lower button counts in the high data regime, while 10-step rollouts generally perform better in the low data regime. Error bars show 95% confidence intervals.

