# OpenReview forum: "The Benefits of Model-Based Generalization in Reinforcement Learning"
_ICLR.cc/2023/Conference — Submitted to ICLR 2023_

### Official Review · Reviewer_4bHT · 2022-10-21

**Confidence:** 4
**Correctness:** 3
**Technical Novelty And Significance:** 2
**Empirical Novelty And Significance:** 3
**Recommendation:** 5

**Clarity, Quality, Novelty And Reproducibility:**

This paper is well-written in general, except for a few ambiguities:
- The second bullet point in Section 3 states that “The return should depend sharply on the policy”, and it’s unclear to me what “sharply” means here. Does it mean that the return of a policy fluctuates a lot? Then does the sparse reward MDP (e.g., the PanFlute environment) satisfy this statement?
- How large is the horizon length in all the environments?

Some references are missing in this paper. E.g., [1] also discusses when model-based methods are better than model-free methods. On a high level, both [1] and this paper constructs environments where the dynamics are simple functions but the corresponding Q function is complicated.

[1] Dong, Kefan, et al. "On the expressivity of neural networks for deep reinforcement learning." International Conference on Machine Learning. PMLR, 2020.


**Strength And Weaknesses:**

Strengths:

- This paper studies the benefit of model-based generalization by comparing model-based rollouts with experience replay. To some extent, this comparison is much fairer because the only difference is the distribution of training data (replay buffer vs model rollouts). This approach is neat and results in some interesting theoretical results (Theorem 1).
- The illustrative example in Section 2 is very easy-to-follow and already well demonstrates the intuitions of this paper.

Weaknesses:

- This paper only discusses one side of the story. In fact, model-based method outperforms experience replay in all the environments studied in this paper. Together with Theorem 1, is it possible to make the claim that model-based methods are always better than experience replay? In other words, discussions and ablation studies about the necessity of the three characteristics in Section 3 are missing. Hence, the scope of the conclusions of this paper is unclear.
- It’s unclear to me when and how the characteristics proposed in Section 3 can be applied to benchmarking environments such as OpenAI Gym. For example, if we predict whether model-based methods are better using the intuition in this paper, does the prediction match reality? Since this is mostly an empirical paper, I would expect more discussions on the practical impact of the conclusions.
- the experiments provide one possible explanation for why MB is better than ER --- learning the transitions can be easier because of the implicit bias of the function class (e.g., the factored structure of the transition) and therefore MB has better sample efficiency. This explanation is somewhat expected and whether it is the main factor in benchmarking experiments such as Mujoco is unclear.


**Summary Of The Paper:**

This paper studies the benefits of model-based generalization by comparing the performance of DQN on the rollouts of a learned model, and DQN on a replay buffer. Theoretically, this paper proves that learning a transition model can be more efficient (in terms of pruning invalid solutions) than learning a Q-function using Bellman equations. Empirically, this paper proposes three environment characteristics where the model-based methods could be better, and demonstrates them using constructed toy environments.

**Summary of the experiments**

Setup:

Experience replay (ER): Run DQN with a replay buffer of the past trajectories (called $\mathcal{D}$)

Model-based: use $\mathcal{D}$ to train a model. And then train a DQN on fake trajectories generated by the learned model starting from a random state from $\mathcal{D}$ (similar to MBPO).

The goal is to make the training of DQN exactly the same except for the data distribution (ER vs. model rollouts).

Results:

This paper compares ER and model-based methods on toy environments in the online RL setting. The number of updates to the Q-network is the same for model-based and ER. In every update, the Q-network is trained using 320 samples from the replay buffer for ER, and 32 model rollouts with length 10 for model-based methods (so the batch size is also the same). This paper shows that for the toy environments model-based is in general better than ER (Figure 3).

**Summary Of The Review:**

On the plus side, the problem setup of this paper is neat and interesting, the results are clean and intuitive. On the minus side, more discussions about the practical relevance and ablation studies are needed since the results are mainly empirical. As a result, I recommend a weak rejection for the current version of this paper.

---

> ### Author Response · Authors · 2022-11-16
> **Response to 4bHT**
>
> Thank you for your time, encouraging feedback and thoughtful questions. Please also see the general response above for clarification on the significance of the environment characteristics and for an additional experiment (https://ibb.co/FbfDCDy) which helps to contextualize the importance of factored structure. We address your other comments below.
>
> **Together with Theorem 1, is it possible to make the claim that model-based methods are always better than experience replay?**: This is a good question. The answer is no because of the assumptions underlying Theorem 1. Most importantly the assumption of realizability (i.e. that the model class actually contains the true model), and the assumption that the value function and model are derived from the same underlying hypothesis class. Realizability is perhaps not so limiting if one is using overparameterized neural networks as function approximators, but would become important in cases where function approximation resources are limited, where learning a large model would be a burden. However, the latter assumption will generally not hold in practice. In a practical architecture, it may be easier to represent certain prior knowledge in a value function than in a model.
>
> It’s easy to come up with cases where, in a specific problem, directly learning a policy (or value function) parameterized in a specific way is better than learning a model parameterized in a specific way. What our result suggests is that in the case where a model and value function have the same prior knowledge of the environment to start with, the model-based approach will generally pull more relevant information from the available data. In other words, when considered over a known class of problems, the model-based approach has an advantage that can generally not be replicated simply by encoding analogous prior knowledge of the problem class into the value function.
>
> Aside from this, there are many other practical considerations such as the computational cost of training and querying a model, which Theorem 1 does not address at all.
>
> **For example, if we predict whether model-based methods are better using the intuition in this paper, does the prediction match reality?**: While this is interesting to consider, our intention was not for the characteristics listed to be inherently predictive of the relative performance of model-free and model-based approaches in general. Please see the general response for further discussion on this point.
>
> **Meaning of “the return should depend sharply on the policy”**: The intended meaning is essentially the same thing as is commonly meant by sparse reward. However, we find the phrase “sparse reward” to be somewhat ambiguous in itself (see [1] for some additional perspective on this point). More precisely, the return depends sharply on the policy if, across the set of all possible policies, the expected return is very similar across most policies and significantly higher only for a relatively small set. For example, in ProcMaze, the reward is always -1 until a goal is reached, technically this is dense in the sense that nonzero reward occurs every step but most policies will fail to reach the goal in reasonable time and thus achieve similarly poor expected return. This motivated our use of the phrase “the return should depend sharply on the policy” instead of, for example,  “the reward should be sparse”.
>
> **How large is the horizon length in all the environments?**: None of our environments are finite horizon in the sense of having a fixed number of steps until termination. ProcMaze is episodic while PanFlute and ButtonGrid are continuing (the agent environment interaction never terminates). However, we do use a discount factor of 0.9 in all our experiments. For evaluation, we also use 100 steps in a newly initialized environment in all cases.
>
> **On the recommended citation**: Thank you for the recommendation, we agree it is relevant due to the shared motivation of highlighting environments where model-based RL is beneficial and will add a citation in the final version of the paper.
>
> **References**
>
> [1]  Arumugam, D., Henderson, P., & Bacon, P. L. (2021). An information-theoretic perspective on credit assignment in reinforcement learning. arXiv preprint arXiv:2103.06224.

---

> > ### Comment · Reviewer_4bHT · 2022-11-25
> > **Follow-up questions**
> >
> > Thank you for the reply and the additional experiments. I still have some follow-up questions listed below:
> >
> > > An important and subtle point that was perhaps underemphasized is that, in Theorem 1, the model-based and model-free approaches begin with the same hypothesis class, and thus the same information about the possible optimal action-value functions.
> >
> > The interpretation of Theorem 1 here seems questionable. Why do model-based and model-free approaches begin with the same hypothesis class? For model-based methods, the hypothesis is the class of transition functions (say, $\mathcal{T}$) and its associated class of optimal action-value function $H_Q$. For model-free methods, the hypothesis is the class of value functions $H_Q$. Hence, the model-based method has *more* prior knowledge about the environment because of the transition class $\mathcal{T}$.
> >
> > To ensure that model-based and model-free approaches have the same prior knowledge, the transition class should be the set of *every* possible transition whose optimal action-value function is in $H_Q$, which is a larger hypothesis class than $\mathcal{T}$. And in this case, $H_M(D)=H_B(D)$ (this is actually the reason why $H_M(D)=H_B(D)$ for tabular MDPs --- the transition class does not provide more prior knowledge).
> >
> > If I understand the interpretation of Theorem 1 correctly (please correct me otherwise), this leads to another follow-up question regarding the contribution/significance of this paper. The experiments provide one possible explanation for why MB is better than ER --- learning the transitions can be easier because of the implicit bias of the function class (e.g., the factored structure of the transition), which is somewhat expected. And this may or may not be the main factor in benchmarking experiments such as Mujoco. In fact, there are other explanations such as a high UTD ratio of the model-based methods [1]. Hence, the contribution of this paper is somewhat unclear.
> >
> > [1] Chen, Xinyue, et al. "Randomized ensembled double q-learning: Learning fast without a model." arXiv preprint arXiv:2101.05982 (2021).

---

> > > ### Author Response · Authors · 2022-11-26
> > > **Reply to Follow-up questions**
> > >
> > > > Why do model-based and model-free approaches begin with the same hypothesis class?
> > >
> > > This is a reasonable question. Our interpretation was that both approaches begin with knowledge of $\mathcal{T}$. By only enforcing Bellman optimality, as in the model-free approach, we are limiting the value-relevant information that can be extracted compared to the model-based approach with the same prior knowledge. We view this as a limitation of the way $\mathcal{T}$ is used by the model-free approach as opposed to a modification of $\mathcal{T}$ itself. We believe this is a reasonable abstraction of model-free learning with bootstrapping on a value function class parameterized to encode the information that the true dynamics belong to $\mathcal{T}$.
> > >
> > > One could indeed enforce that the model-based approach uses the hypothesis class consisting of all transition functions consistent with $H_Q$. In that case (as in our tabular result) the two approaches would be equivalent. However, unlike the scenario we consider, this is not an inherent limitation of the model-based approach but an arbitrary restriction. Moreover, even under this restriction, the model-based approach is still not worse, but merely equivalent.
> > >
> > > We believe that Theorem 1 will be of interest to many in the community because it contradicts a reasonable intuition.  One may think that by parameterizing a value function in a way that enforces some known structure of the environment, one could obtain the same generalization benefits as the model-based approach. However, Theorem 1 suggests that this is not the case.
> > >
> > > > In fact, there are other explanations such as a high UTD ratio of the model-based methods [1].
> > >
> > > The update-to-data (UTD) ratio was one of the things we controlled between the model-based and model-free approaches in our experiments. In particular, all approaches use the same number of value function updates per environment step (1 in the high data regime and 10 in the low data regime) as well as the same number of transitions (320) in each update. The only difference is whether those transitions are pulled from a replay buffer or generated by a learned model.
> > >
> > > > Hence, the contribution of this paper is somewhat unclear.
> > >
> > > Our main empirical contribution is to demonstrate environments where the ability of model-based learning to extract more value-relevant information from the data leads to a significant benefit. We focused on environments with factored structure to ensure that there is a lot of information in the data that the model-free approach may fail to capitalize on. We showed that model-based approaches display a significant benefit in these environments as environment complexity is increased. We also believe that some form of factored structure is likely to be present in many applications of practical interest and hence worthwhile to consider in its own right.

---

### Official Review · Reviewer_8CUu · 2022-10-24

**Confidence:** 5
**Correctness:** 2
**Technical Novelty And Significance:** 2
**Empirical Novelty And Significance:** 3
**Recommendation:** 5

**Clarity, Quality, Novelty And Reproducibility:**

The clarity is good. The paper will benefits from more thorough theoretical analysis (see above section for detailed comments). The empirical results should be reproducible.

**Strength And Weaknesses:**

Strength:
- An interesting and important question to study, as when and why model-based RL is guaranteed to be better than experience replay can significantly influence us to design practical RL algorithms for real-world applications
- Well-written and easy to follow, with many concrete examples and interesting discussions of empirical observations
- Good introduction of the background

Weaknesses/Questions:
- The first motivating example in the introduction (an agent trying to cross a stream to reach the food) is not very clear and convincing. It is hard to understand the formulation of the state (is it a bandit problem given your state definition?), and what is the data (some state+reward pairs?) in your experience replay, and why model-free learning clearly/definitely fails (it can also generalize via value function)?
- Perhaps the main rigorous answer is Theorem 1, where I was expecting to see when and why model-based RL is better. However, I found the result is not surprising/contains enough theoretical insights. It is basically saying, if we are given a set of hypothesis class for the transition dynamics, and we can successfully identify one that perfectly matches the data, then using that transition function (probably optimal if data is enough) to solve Bellman optimal equation is better than directly using the data to solve. This might be trivial and in practice, the key question is actually what theoretical guarantee we can achieve when the approximate model makes an error (upper bounded by some constant). If the data D is very limited and the best model consistent with D is still far from the true dynamics, then is model-based RL always better (since it incurs compounding error)? Another essential question to study is how will the model error propagates to solving Bellman optimality equations (the data is always from true dynamics but generated ones can be wrong)?
- It seems most above important questions are not rigorously answered. The following sections provide nice examples and hypothesis on the conditions. But I feel they should be an empirical evidence to some more general theorems. I was expecting a formal sufficient (maybe not necessary) and hopefully general enough condition for when model generalization + compounding error is better than ER.
Theoretically this should depend on specific algorithm, data collecting policy and sample size, etc. Otherwise, it may be hard to summarize some general principles that guides the design of algorithm for more complicated and realistic environments.
- The observation that sometimes learned model is even better than the true model is very interesting. In principle, even using more smooth model can help exploration or optimization, this should eventually lead to bias. Will this be an issue and can we alleviate it using annealing (gradually reduce to true model)? In case of no bias, it seems to be something like a reward shaping or simply a coincidence that two MDPs have same solution. This may deserve more thorough study.
- "factored structure" appears many times but seems fuzzy. Can you give a formal definition?

**Summary Of The Paper:**

The paper investigates the question of when and why model generalization is better than value function generalization in reinforcement learning. The paper starts with a theoretical result that gives the intuition on this question: reinforcement learning with a learned model of the transition dynamics can reduce the optimal action value function space than simply solving bellman constraints with experience replay. Then the authors provide some hypothesis/conditions that tends to favor model-based RL, and then justify them by testing on some designed toy experiments.

**Summary Of The Review:**

Interesting question, nice examples and discussion of empirical observation. Needs more thorough and rigorous theoretical study on some key questions.

---

> ### Author Response · Authors · 2022-11-16
> **Response to 8CUu**
>
> Thank you for your time and your appreciation of the question we address and the examples we present. We address most of your specific questions below. Please also see the general response above for clarification regarding the significance of Theorem 1 and how it relates to the proposed environment characteristics.
>
> **Regarding the first motivating example in the introduction**: This was only a motivating example, meant to stimulate thinking about the question we intended to answer. However, we can understand how it might also lead to confusion. The idea is that since the agent has never gained any benefit from pushing a tree over in the past, an approach simply bootstrapping from experience would not have increased the value for that action in any state. Nonetheless, the agent's experience contains information about the generic consequences of pushing a tree over as well as the idea of a fallen tree being useful in certain situations. Using a model, the agent could imagine tieing these two pieces of information together and thus realize the value of pushing over a tree in certain situations. We do not contest that with a sufficiently contrived function approximation scheme this conclusion would not hold. The example problem in the proof of Theorem 1 in the appendix (Figure 5) presents a similarly simple situation in which the benefit of the model-based approach is rigorously worked out.
>
> **Regarding the limitations of Theorem 1**: You are correct that Theorem 1 does not answer the theoretical questions you posed. We nonetheless feel it goes a long way toward motivating the model-based approach to RL in general by clearly showing that the model-based approach can capitalize on information that the model-free approach will fail to use even when the two approaches begin with the same prior knowledge. While the more algorithm-specific questions you posed are interesting, we feel that attempting to address them here might distract from our main conclusions. These questions are better addressed elsewhere, see for example works that provide bounds on the error of the action value function obtained from planning with an inaccurate dynamics model [1,2], or works that develop and analyse algorithms for model-based RL without assuming model realizability [3, 4].
>
> **On the observation that sometimes the learned model is even better than the true model**: We agree that this is interesting. It is not an effect we sought out but rather a surprise that occurred in our experiments that we took additional steps to understand better as we thought it might be of interest to others. The idea of annealed model smoothing to mitigate bias while driving exploration is an interesting direction for future work. We agree this effect deserves further study.
>
> **On the formal definition of factored structure** : By factored structure, we meant that each environment state $S_t$ can be represented as a vector of $n$ components $S_t^i$ for $i\in[0..n-1]$ such that $S_t=(S_t^0,S_t^1,...,S_t^{n-1})$. Furthermore, the state transition dynamics factor as $P(S_{t+1}|S_{t},A_{t})=\prod_iP(S_{t+1}^i|Par(i)\_{t+1},A_{t})$ where $Par(i)\_{t+1}\subseteq \\{S_t^0,S_t^1,...,S_t^{n-1}\\}$. Or in words, the transition dynamics for each component are independent and depend on only a subset of the components at the previous time-step. When Par(i) is small, this can be much simpler to learn than an arbitrary model. All the environments we consider admit a representation with a small Par(i). For example in ProcMaze, the presence of an agent in a given cell at the next step depends only on the action, whether there is currently an agent in the neighbouring cells and whether there is a wall in the same cell.
>
> **References**
>
> [1] Talvitie, E. (2017, February). Self-correcting models for model-based reinforcement learning. In Thirty-First AAAI Conference on Artificial Intelligence.
>
> [2] Talvitie, E. (2018). Learning the reward function for a misspecified model. In International Conference on Machine Learning.
>
> [3] Szita, I., & Szepesvári, C. (2011). Agnostic KWIK learning and efficient approximate reinforcement learning. In 24th Annual Conference on Learning Theory.
>
> [4] Ross, S., & Bagnell, J. A. (2012). Agnostic system identification for model-based reinforcement learning. arXiv preprint arXiv:1203.1007.

---

### Official Review · Reviewer_JiYh · 2022-10-25

**Confidence:** 4
**Correctness:** 4
**Technical Novelty And Significance:** 3
**Empirical Novelty And Significance:** 2
**Recommendation:** 6

**Clarity, Quality, Novelty And Reproducibility:**

The paper is overall well-written and clear. While model-based RL has been well-studied in the literature, the generalization of learned models to new observations has received less attention. I have not seen the paper’s main theorem elsewhere in the literature.

**Strength And Weaknesses:**

### Strengths

- The paper provides nice intuitive examples of generalization in maze environments that highlight the limitations of model-based and value-based RL algorithms.
- The paper is well-written and does a good job at summarizing and emphasizing the key take-aways from each section.
- The experiment setup is clear.
The analysis of different learned models was enlightening as it highlighted the importance of the inductive bias of the world model, and because it provided a striking but also simple setting where a learning algorithm benefits more from an imperfect learned model than from the perfect world model. I appreciated that the authors went further to validate their hypothesis that the learned models were benefitting from reward smoothing in these contexts
- Theorem 1 provides a nice contrast to the findings of Parr et al. in the linear setting.

### Weaknesses

- The paper focuses on a handful of small toy environments. While this is useful to guide the reader’s intuition and to make the experimental findings interpretable, it also limits the generality of the paper’s conclusions.
- The degree to which the findings in this paper also apply to larger scale models is an open question. While the toy settings considered in the paper do a good job of illustrating some environment qualities can than influence the efficacy of simple model-based algorithms, it’s not clear to me that these are the only parameters that can influence the relative performance of model-learning methods in more challenging tasks. For example, the influence of model learning on exploration, or on the stability of the underlying learning method, could also play a role in an agent’s final performance in these settings. It’s not clear to me whether the environment features and model smoothing properties identified in this paper would be a dominant factor in performance compared to these other factors.

- I suspect that the value-based agents might be worse at generalization than policy-based agents based on prior literature on e.g. the ProcGen suite (Raileanu et al., 2020), and would have liked to see the inclusion of a policy-gradient method in the set of baselines.
- The presentation of the proof of theorem 1 could be improved: the definition of the transition dynamics for $S_t[0]$ for example, seem to be defining different quantities in the top and bottom line of the bracket, and the hypothesis class $H$ from which the q-functions are selected is not defined. I assume it is the set of all real-valued functions over the state space, but this should be stated explicitly.
- While Theorem 1 is a nice observation, the statement itself is relatively weak. It does not, for example, provide a means of quantifying the degree to which the hypothesis space can be reduced by model classes with certain properties. Nor does it address settings where the algorithm may need to trade off between a limited set of environment steps and the approximation error of a model. This limits the significance of the theoretical contribution of the paper.


**Summary Of The Paper:**

This paper provides a high-level characterization of the properties of RL tasks in which model-based algorithms can be expected to improve generalization to new observations. Two primary quantities arise from this analysis: first, the state space should exhibit a simple underlying factored structure which the inductive bias of the model is well-suited to learning; second, the distribution of rewarding trajectories that an agent sees should be sufficiently sparse as to make learning from Bellman backups difficult, but still provide non-zero reward density so that there is some signal for a learned model to leverage.

**Summary Of The Review:**

Overall, this paper provides a nice set of interpretable examples that highlight the different behaviour of model-free and different model-learning algorithms. One of these examples in particular illustrates how the smoothness bias in DNN function approximators can facilitate learning and generalization in model-based algorithms. The paper suffers two main limitations: first, its main theoretical result is a relatively weak statement, and doesn’t directly offer a concrete set of implications for the effect of model-learning methods on generalization. Second, it is not clear whether the empirical findings concerning the behaviour of model-based and model-free methods will generalize outside of the small environments presented in the paper.

---

> ### Author Response · Authors · 2022-11-16
> **Response to JiYh**
>
> Thank you for your time and thoughtful feedback. Please see the general response above for the rationale behind our choice of environments and for some clarification on the significance of Theorem 1 and the environment characteristics in Section 3. We address your more specific questions below.
>
> **Regarding including a policy-based agent**: While comparing the generalization performance of policy-based methods and value-based methods is interesting we feel it is beyond the scope of this paper. We chose to use Q-learning as the underlying behaviour-learning algorithm in all cases because it is naturally off-policy and thus easy to apply with either data from a replay buffer or a learned model, allowing for a straightforward, direct, comparison of these two data sources. Correctly applying a policy-based method like actor-critic with a replay buffer would require off-policy correction, usually via importance sampling, which would introduce many confounding factors which we wished to avoid.
>
> **It’s not clear to me whether the environment features and model smoothing properties identified in this paper would be a dominant factor in performance**: Please see the new result here: https://ibb.co/FbfDCDy (explained in the general response above) which considers an environment without factored structure where the model-based approach becomes worse with increasing complexity. This example helps to contextualize our positive results for model-based learning by showing that the model-based approach underperforms in an environment with unstructured transitions. More thoroughly investigating the relative importance of different factors is an interesting direction for future work.
>
> **Regarding the definition of the transition dynamics for the example in the proof of theorem 1**: There was an error in the definition of the transition dynamics, the expression should read:
> \begin{align*}
> S_{t+1}[0]&=\text{$S_t[0]-1$ if $S_t[0]>0$}\\\\
> S_{t+1} &=\text{$\bot$ otherwise}
> \end{align*}
> Thank you for pointing this out! We will fix this in the final paper.
>
> **Regarding the definition of the hypothesis class H in Theorem 1**: As far as we can see, all hypothesis classes are well-defined, please see the explanation below for details and let us know if this does not address your question.
>
> The hypothesis class H is not over q-functions but transition functions, it is defined to be an arbitrary class of deterministic transition functions, and the theorem holds for all such hypothesis classes. The hypothesis class $H_Q$ which is over q-functions is defined in terms of H as
> $H_Q=\\{q:s,a\rightarrow \mathbb{R}|\exists p\in H, \forall s,a\ q(s,a)=r(s,a)+\max_{a'} q(p(s,a),a')\\}$
>
> in words, this is simply the class of optimal action values for MDPs with fixed reward function $r$ and transition dynamics belonging to $H$. This is in general smaller than the set of all real-valued functions over state-action space which is important because this means the class of possible value functions prior to observing any data is the same for both the model-based and model-free approaches.
>
> The fact that the hypothesis class over q-functions is limited in this way makes the result more interesting because the two approaches begin with the same information, so the model-free approach is actually failing to make use of information about the value function which is available in the data. If instead the initial hypothesis class for the model-free method was the set of all real-valued functions, one could argue that the model-free method simply started with a larger hypothesis class, making the result less interesting. In that case, the model-based approach could result in a smaller set of possible optimal action-value functions even in the tabular case simply because it started with a more restricted hypothesis class.

---

### Official Review · Reviewer_5uCS · 2022-11-01

**Confidence:** 4
**Correctness:** 4
**Technical Novelty And Significance:** 4
**Empirical Novelty And Significance:** 4
**Recommendation:** 8

**Clarity, Quality, Novelty And Reproducibility:**

The paper is well written. The contributions are well motivated and, to the best of my knowledge, novel.

It might be worth mentioning [1] in the related work (e.g., as part of paragraph 3, p. 9). Although that work doesn't consider generalization, it does compare the gradient dynamics and convergence rates when using "end-to-end" model-based method to when the value function is encoded directly with an equally expressive estimator.

[1] Clement Gehring, Kenji Kawaguchi, Jiaoyang Huang, Leslie Pack Kaelbling. Understanding End-to-End Model-Based Reinforcement Learning Methods as Implicit Parameterization. NeurIPS, 2021

**Strength And Weaknesses:**

This work examines an important open question about the benefits of model-based RL. Although these methods are commonly believed to be more sample efficient, formal results on the subject are sparse. This work will likely be of interest to the RL community.

**Summary Of The Paper:**

This work formalizes the generalization benefits of model-based RL methods in the context of deterministic episodic MDPs who's transitions probability belong to a known hypothesis class. The authors then discuss the intuition gained from their theory in various illustrative environments and take a closer look at the case when transitions can be assumed to be factored.

**Summary Of The Review:**

Assuming the novelty isn't brought in question, this paper seems like a clear accept. Ideas are well motivated and of interest to RL.

---

> ### Author Response · Authors · 2022-11-16
> **Response to 5uCS**
>
> Thank you for the encouraging feedback and for recommending the additional citation. We agree that it is relevant and will add it to our discussion of implicit models in the related work as suggested.

---

### Author Response · Authors · 2022-11-16
**General Response**

We thank the reviewers for their time and encouraging feedback. We address some general points here and reply to each reviewer's specific questions below the associated review. We have also run an additional experiment in an environment without factored structure to better contextualize our positive results for model-based learning. The results can be found here: https://ibb.co/FbfDCDy (see below for a complete description).

**Regarding the significance of Theorem 1**: As mentioned by reviewers JiYh and 8CUu,  Theorem 1 is indeed a fairly simple result. Nonetheless, we feel it is apriori nontrivial and provides strong motivation for the benefit of model-based reinforcement learning in general. For these reasons, we see simplicity as an asset as this makes it accessible and broadly relevant.

An important and subtle point that was perhaps underemphasized is that, in Theorem 1, the model-based and model-free approaches begin with the same hypothesis class, and thus the same information about the possible optimal action-value functions. Thus, it is not that the model-based approach simply has more prior knowledge to begin with. Rather, both approaches begin with this knowledge but the model-free approach fails to extract some value-relevant information from the data. This is especially interesting because we show the approaches are equivalent in the tabular case, and prior work suggests they are also equivalent for policy evaluation in the linear case, thus using intuition from these cases will lead to the wrong conclusion in general.

**Regarding the necessity of the characteristics listed in Section 3**: This was mentioned by Reviewers JiYh and 4bHT.  We do not claim that these characteristics are necessary conditions for model-based approaches to outperform model-free approaches. Rather they served as guidelines for us to suggest domains in which we expected the benefits of model-based approaches to be most prominent. These guidelines were motivated by our theoretical observation that the model-free approach generally loses value-relevant information from the data. By considering environments with factored structure, we ensure that there is a lot of information in the data that the model-free approach may fail to capitalize on (factored structure is not the only way to ensure this). By considering environments where the return depends sharply on the policy (roughly speaking where the reward is sparse, see response to 4bHT), we aim to make it hard to learn a good policy without exploiting the factored transition dynamics. Finally, we ensure that the agent can actually observe rewards since otherwise even the model-based approach would be hopeless, as no meaningful value learning could occur either inside or outside the model.

We also performed an additional experiment in an environment which lacks factored structure, outlined below. We emphasize that this is not intended as a conclusive ablation study for the necessity of the characteristics we outline. However, it does help to better contextualize our positive results for model-based learning.

**Regarding the simplicity of the environments in the experiments**: This was mentioned by Reviewers JiYh and 4bHT. While these environments are simple in some ways, they are designed to be challenging in the sense that they have combinatorial dynamics which must be exploited to succeed. Learning to navigate in randomized mazes for example is much harder than learning to perform well in a fixed maze environment as is often considered. The literature is already rich with examples of the capabilities of various model-based agents in larger-scale domains (for example [1]). Here we focus on developing a better understanding of some basic reasons these methods can improve upon model-free learning. We feel this is better done in carefully chosen domains to start with, where interpretability is not impeded. Evaluating the extent to which the conclusions carry over to larger-scale domains is an interesting direction for future work.

**Update**: We have updated the paper to include the new experiment, the suggested citations, and some other minor changes.

**References**

[1] Hafner, D., Lillicrap, T., Norouzi, M., & Ba, J. (2020). Mastering atari with discrete world models. arXiv preprint arXiv:2010.02193.

---

> ### Author Response · Authors · 2022-11-16
> **New Experiment**
>
> **New experiment on an environment which lacks factored structure**: As mentioned by 4bHT, our paper primarily focuses on highlighting environments in which model-based learning is expected to be beneficial. Nevertheless, we agree it is worthwhile to contrast this with what happens in environments which do not have such favourable characteristics. To that end, we ran an additional experiment on an environment without factored structure, the results can be found at the anonymous link here: https://ibb.co/FbfDCDy.
>
> The environment in this case was simply an open grid with a goal in the bottom right corner and a reward of -1 for every step until the goal is reached at which point termination occurs. The agent starts in a random location in each episode. As in our other experiments, we include occasional spontaneous transitions to the goal. The agent location is simply represented by a one-hot vector (effectively tabular) so there is really no structure to exploit, the learned model must essentially memorize every individual transition to learn the dynamics.
>
> Our experimental design was the same as in Section 4. We tune the Q-network step size and softmax exploration temperature from the same set of values on a grid size of size 12 and then used the best hyperparameters for each agent on the other grid sizes. In this experiment, we focused on the low-data regime where the simple model tended to have the biggest advantage over ER in our other experiments.
>
> In contrast to our other experiments, here we see that the simple model becomes worse relative to ER as the environment complexity increases. This is reasonable as the model has no ability to extrapolate beyond the data, the best it can do is memorize what is already in the replay buffer and the limitations of finite model capacity and imperfect optimization prevent it from doing so perfectly. We plan to include this experiment in an appendix of our final paper to contextualize our other results.

---

### Decision · Program_Chairs · 2023-01-20

**Decision:**

Reject

**Justification For Why Not Higher Score:**

In the AC's opinion, the main weakness of the paper is that "the result is not surprising/contains enough theoretical insights". I copied reviewer 8CUu's second bullet here for more detail.

"It is basically saying, if we are given a set of hypothesis class for the transition dynamics, and we can successfully identify one that perfectly matches the data, then using that transition function (probably optimal if data is enough) to solve Bellman optimal equation is better than directly using the data to solve. This might be trivial and in practice, the key question is actually what theoretical guarantee we can achieve when the approximate model makes an error (upper bounded by some constant). If the data D is very limited and the best model consistent with D is still far from the true dynamics, then is model-based RL always better (since it incurs compounding error)? Another essential question to study is how will the model error propagates to solving Bellman optimality equations (the data is always from true dynamics but generated ones can be wrong)?"

The AC had an extensive discussion with the reviewer who gave a rating 8, on this topic as well.

Reviewer 5uCS says that “Thm. 1 establish a simple setting where no matter what, we cannot leverage structured transitions by enforcing the Bellman equations despite being able to perfectly leverage all information in the data.” and “This provides some surprisingly clear yet subtle insight about the decades old debate surrounding the model-free vs. model-based.”

The AC didn’t agree with Reviewer 5uCS and agrees more with the other reviewers. The AC pointed out a 2019 paper  https://arxiv.org/pdf/1811.08540.pdf, which demonstrates the same mechanism with a much more quantitative approach. The paper linked above actually implies Theorem 1 (but it did much more than what Theorem 1 of this paper offers). In more accessible language, the paper above essentially says that if your model is a sparse reward problem, then fitting the Q doesn't help, but the dynamics can extrapolate. In the language of this paper under review, I think it says that if you have a sparse reward problem, and say the sparse reward is somewhere very far away from the origin, and you have some replay buffer that only has tuples near the origin, then fitting Q on this reply buffer tells you very little (at least nothing about dynamics because whatever dynamics will induce the same Q as long as the reward is zero around the origin), but fitting the dynamics will tell you much more the dynamics at places far away from the origin (assuming the dynamics extrapolate.). So this would imply bullet 2 of theorem 1.

Therefore, the AC thinks the theoretical value of the paper is limited and does not meet the bar of acceptance for neurips.



**Justification For Why Not Lower Score:**

N/A

**Metareview: Summary, Strengths And Weaknesses:**

In the AC's opinion, the main weakness of the paper is that "the result is not surprising/contains enough theoretical insights". I copied reviewer 8CUu's second bullet here for more detail.

"It is basically saying, if we are given a set of hypothesis class for the transition dynamics, and we can successfully identify one that perfectly matches the data, then using that transition function (probably optimal if data is enough) to solve Bellman optimal equation is better than directly using the data to solve. This might be trivial and in practice, the key question is actually what theoretical guarantee we can achieve when the approximate model makes an error (upper bounded by some constant). If the data D is very limited and the best model consistent with D is still far from the true dynamics, then is model-based RL always better (since it incurs compounding error)? Another essential question to study is how will the model error propagates to solving Bellman optimality equations (the data is always from true dynamics but generated ones can be wrong)?"

The AC had an extensive discussion with the reviewer who gave a rating 8, on this topic as well.

Reviewer 5uCS says that “Thm. 1 establish a simple setting where no matter what, we cannot leverage structured transitions by enforcing the Bellman equations despite being able to perfectly leverage all information in the data.” and “This provides some surprisingly clear yet subtle insight about the decades old debate surrounding the model-free vs. model-based.”

The AC didn’t agree with Reviewer 5uCS and agrees more with the other reviewers. The AC pointed out a 2019 paper  https://arxiv.org/pdf/1811.08540.pdf, which demonstrates the same mechanism with a much more quantitative approach. The paper linked above actually implies Theorem 1 (but it did much more than what Theorem 1 of this paper offers). In more accessible language, the paper above essentially says that if your model is a sparse reward problem, then fitting the Q doesn't help, but the dynamics can extrapolate. In the language of this paper under review, I think it says that if you have a sparse reward problem, and say the sparse reward is somewhere very far away from the origin, and you have some replay buffer that only has tuples near the origin, then fitting Q on this reply buffer tells you very little (at least nothing about dynamics because whatever dynamics will induce the same Q as long as the reward is zero around the origin), but fitting the dynamics will tell you much more the dynamics at places far away from the origin (assuming the dynamics extrapolate.). So this would imply bullet 2 of theorem 1.

Therefore, the AC thinks the theoretical value of the paper is limited and does not meet the bar of acceptance for neurips.



**Summary Of Ac-Reviewer Meeting:**

The AC asks for scheduling meetings, but none of the reviewers replied to the scheduling request.